# Aridification signatures from fossil pollen indicate a drying climate in east-central Tibet during the late Eocene

Qin Yuan[a,b,c,d*], Natasha Barbolini[e,f], Catarina Rydin[e,g], Dong-Lin Gao[a,b], Hai-Cheng Wei[a,b], Qi-Shun Fan[a,b], Zhan-Jie Qin[a,b], Yong-Sheng Du[a,b], Jun-Jie Shan[a,b,c], Fa-Shou Shan[a,b], Vivi Vajda[d]

[a] *Key Laboratory of Comprehensive and Highly Efficient Utilization of Salt Lake Resources, Qinghai Institute of Salt Lakes, Chinese Academy of Sciences, Xining, China*

[b] *Qinghai Provincial Key Laboratory of Geology and Environment of Salt Lakes, Qinghai Institute of Salt Lakes, Chinese Academy of Sciences, Xining, China*

[c] *University of Chinese Academy of Sciences, Beijing 100049, China*

[d] *Department of Palaeobiology, Swedish Museum of Natural History, Stockholm, Sweden*

[e] *Department of Ecology, Environment and Plant Sciences and Bolin Centre for Climate Research, Stockholm University, SE-106 91 Stockholm, Sweden*

[f] *Department of Ecosystem and Landscape Dynamics, Institute for Biodiversity and Ecosystem Dynamics, University of Amsterdam, 1098 XH The Netherlands*

[g] *The Bergius Foundation, The Royal Swedish Academy of Sciences, Box 50005, SE-104 05 Stockholm, Sweden*

*Correspondence to*: Natasha Barbolini (barbolini.natasha@gmail.com)

**Abstract.** Central Asia experienced a number of significant elevational and climatic changes during the Cenozoic, but much remains to be understood regarding the timing and driving mechanisms of these changes, as well as their influence on ancient ecosystems. Here we describe the palaeoecology and palaeoclimate of a new section from the Nangqian Basin in Tibet, northwestern China, here dated as Bartonian (41.2–37.8 Ma; late Eocene) based on our palynological analyses. Located on the east-central part of what is today the Tibetan Plateau, this section is excellently placed for better understanding the palaeoecological history of Tibet following the India-Asia collision. Our new palynological record reveals that a strongly seasonal steppe-desert ecosystem characterised by drought-tolerant shrubs, diverse ferns and an underlying component of broad-leaved forests existed in east-central Tibet during the Eocene, influenced by a southern monsoon. A transient warming event, possibly the Middle Eocene Climatic Optimum (MECO; 40 Ma), is reflected in our record by a temporary increase in regional tropical taxa and a concurrent decrease in steppe-desert vegetation. In the late Eocene, a

drying signature in the palynological record is linked to proto-Paratethys sea retreat, which caused widespread

long-term aridification across the region. To better distinguish between local climatic variation and farther-

reaching drivers of Central Asian palaeoclimate and elevation, we correlated key palynological sections across

the Tibetan Plateau by means of established radioisotopic ages and biostratigraphy. This new palynozonation

illustrates both intra- and inter-basinal floral response to Qinghai-Tibetan uplift and global climate change

during the Paleogene, and provides a framework for the age assignment of future palynological studies in

Central Asia. Our work highlights the ongoing challenge of integrating various deep time records for the

purpose of reconstructing palaeoelevation, indicating that a multiproxy approach is vital for unravelling the

complex uplift history of Tibet and its resulting influence on Asian climate.

## 1. Introduction

A series of major geological events occurred during the Cenozoic, which led to a fundamental change in

the global climate (Zachos et al., 2001). The most important events include the formation of the polar ice cap

(e.g., DeConto and Pollard, 2003; Pagani et al., 2011), regression of the proto-Paratethys Sea from Eurasia

(Abels et al., 2011; Bosboom et al., 2014; Caves et al., 2015; Bougeois et al., 2018; Kaya et al., 2019; Meijer et

al., 2019), and uplift of the Qinghai-Tibetan region (Dupont-Nivet et al., 2007, 2008; Molnar et al., 2010; Miao

et al., 2012; Hu et al., 2016; Li et al., 2018). Today the Tibetan Plateau (TP) is the highest elevated plateau in the

world, with a complex uplift history beyond a simple collision between the Indian and Asian continents (Molnar

and Tapponnier, 1975; Aitchison and Davis, 2001; Wang, C.S., et al., 2008; Xia et al., 2011; Aitchison et al.,

2011; Zhang et al., 2012; Wang, C.W., 2014; Spicer et al., 2020). Here, the term 'Tibetan Plateau' is used in the

paper to denote the geographic extent occupied by the modern plateau, but should not be taken to imply that an

elevated expanse of low relief topography existed across this region in the Eocene (Spicer et al., 2020).

Previous studies indicate that retreat of the proto-Paratethys Sea and the uplift of Tibet as well as other

ranges to the north, such as the Altai, Sayan, and Hangay (Caves et al., 2014), may have been responsible for

monsoon intensification and aridification across the Asian continental interior in the Paleogene, although the

timing of these mechanisms, and their roles in forcing climate dynamics, are still debated (Caves et al., 2015;

Spicer, 2017). In particular, a lack of consensus exists regarding the onset of Asian aridification, whether it was

a Paleogene or Neogene phenomenon, and its relationship with Tibetan uplift (e.g., Dupont-Nivet et al. 2007;

Xiao et al., 2010; Miao et al., 2012; Caves et al., 2015; Liu et al., 2016; Wang et al., 2018; Li L. et al., 2019;

Paeth et al., 2019). Aridification in northeastern Tibet appears to have intensified after the Middle Eocene
Climatic Optimum (MECO;~ 40 Ma), a short-lived warming event documented in marine records globally. The
drying climate after this event is primarily linked to the second regression of the proto-Paratethys Sea, which
reduced moisture supply via the westerlies to Central Asia (Kaya et al., 2019). In northeastern Tibet, the regional
disappearance of perennial lakes, accompanied by an increase in pollen from xerophytic plants, marks a
permanent aridification step in the Asian terrestrial record after ~40 Ma (Bosboom et al., 2014); however, these
climatic trends are yet to be identified in central Tibet.

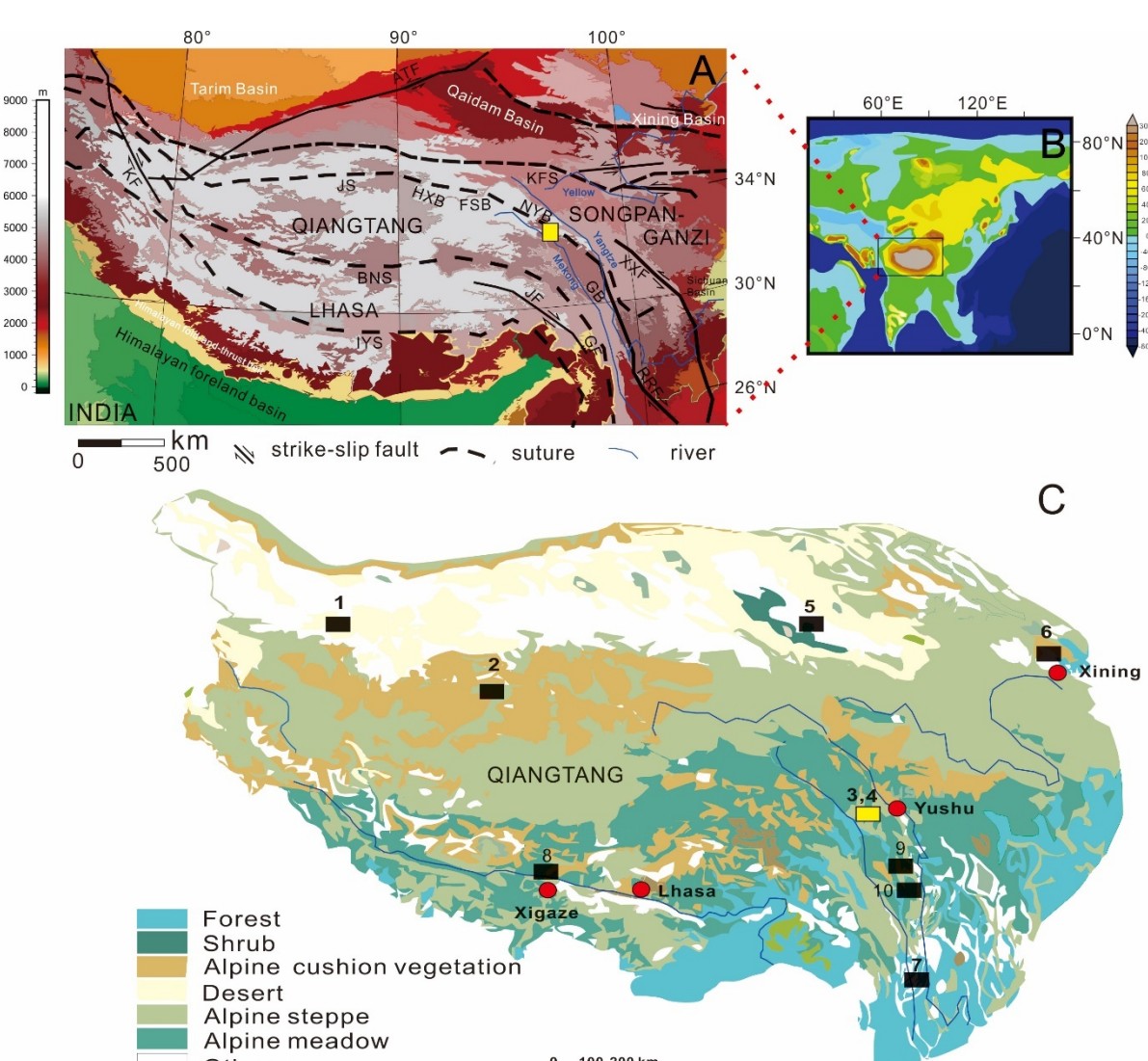


**Figure 1: (A) Tectonic map of the Tibetan Plateau (TP) with major sedimentary basins (HXB: Hoh Xil Basin; FSB:**
**Fenghuo Shan basins; NYB: Nangqian-Yushu basins; GB: Gongjo Basin), sutures (JS: Jinsha suture; BNS: Bangong-**
**Nujiang suture; IYS: Indus-Yalu suture), and major faults (KF: Karakorum fault; ATF: Altyn Tagh fault; KFS:**
**Kunlun fault system; XXF: Xiangshuihe-Xiaojiang fault system; RRF: Red River fault; GF: Gaoligong fault; JF:**
**Jiali fault) indicated, redrawn after Horton et al. (2002). The yellow rectangle indicates the location of this study in**

the Nangqian Basin. (B) late–middle Eocene (40 Ma) palaeogeographic reconstruction with the Qinghai-Tibetan

region indicated by a black rectangle (redrawn after Tardif et al., 2020). (C) Modern vegetation distributions on the

Tibetan Plateau with major towns indicated in red (redrawn after Baumann et al., 2009). Numbers indicate the

positions of palynological assemblages that are correlated in Fig. 4 and the text: 1. Tarim Basin; 2. Hoh Xil Basin; 3,

4. Nangqian Basin (this study indicated by a yellow rectangle); 5. Qaidam Basin; 6. Xining Basin; 7. Jianchuan Basin;

8. Xigaze Basin; 9. Markam Basin; 10. Gonjo Basin.

The uplifting, large-scale thrusting and striking of Tibet caused several Paleogene intracontinental basins to
form within the northern and central Qinghai-Tibetan region, including the Nangqian Basin. Situated in the
Yushu area (Fig. 1), this basin lies directly above the Lhasa terrane, which comprised part of NE Gondwana in
the Late Triassic to Early Jurassic and formed through a subduction–accretion process similar to that of the later
India–Asia collision (Liu et al., 2009). Subsequent to its formation, the Nangqian Basin was infilled with non-
marine sedimentary deposits (Wang et al., 2001; 2002), and is now a key site for understanding the Cenozoic
tectonics, palaeoelevation and paleoclimatic changes that took place in the Qinghai-Tibetan region since the
collision of the Indian and Asian tectonic plates (Gupta et al., 2004; Molnar, et al., 2004; Wang et al., 2001).
Previous palynological studies from this part of the plateau revealed a relatively dry climate with brief humid
intervals in the late Eocene, dominated by drought-tolerant (xerophytic) and salt-tolerant (halophytic) steppe-
desert vegetation (Wei, 1985; Yuan et al., 2017).
This climate and palaeoflora were very similar to contemporaneous plateau ecosystems further to the north,
such as the Xining (Dupont-Nivet et al. 2007, 2008; Hoorn et al., 2012) and Hoh Xil (Liu et al., 2003; Miao et
al., 2016) basins, demonstrating the potential for these successions to be biostratigraphically correlated.
Furthermore, oxygen isotope records indicate that both northern and east-central Tibet received moisture
dominantly via the westerlies, which have maintained a semi-arid to arid climate in Central Asia since the early
Eocene (Caves et al., 2015; Caves Rugenstein and Chamberlain, 2018. This suggests that aridification across
this part of Tibet in the Eocene was related to large-scale atmospheric transport, and justifies a comparison of
palynological records in the northern and central parts of the TP.
In contrast, southeastern Tibet seems to have experienced a more humid climate hosting widespread
conifer and warm-temperate broad-leaved forests (Li et al., 2008; Su et al., 2018), likely influenced by a
Paleogene Inter-tropical Convergence Zone (ITCZ) -driven monsoon system similar to the modern Indonesia-
Australia Monsoon (I-AM; Spicer et al., 2017). Today this summer-wet, winter-dry monsoonal regime presides
over a biodiversity hotspot in southern Asia; similarly seasonal climates in the past are thought to also have
stimulated high biodiversity (Spicer, 2017). Southerly moisture has probably rarely extended northward of the
central TP (Caves Rugenstein and Chamberlain, 2018); moreover, southern Tibetan Eocene floras display a
modern aspect (e.g., Linnemann et al., 2018) that is quite different to more ancestral steppe vegetation hosted in
the northern TP.

The extent and timing of mechanisms that promoted somewhat different floras south and north of the

Tibetan–Himalayan orogen remain poorly understood, with Licht et al. (2014) reporting marked monsoon-like
patterns in both regions during the Eocene, utilising records from northwest China and Myanmar. The role of
Qinghai-Tibetan uplift also remains unclear, with contrasting models of plateau evolution supported by various
tectonic, isotopic, modelling, and biological evidence (e.g., Mulch and Chamberlain, 2006; Rowley and Currie,
2006; Ding et al., 2014; Li et al., 2015; Jin et al., 2018; Botsyun et al., 2019; Su et al., 2019; Valdes et al., 2019;
Shen and Poulsen, 2019 and see summaries in Spurlin et al., 2005; Wang et al., 2014; Spicer, 2017).
Accordingly, further stratigraphic and paleoenvironmental studies of the sedimentary successions within these
basins are necessary to provide clarification on local vs. regional climatic changes experienced as a result of
uplift, global cooling, and progressive aridification in Central Asia during the Paleogene.

The location of the Nangqian Basin on the east-central part of the TP provides an ideal locality for testing

the influence of these mechanisms on Asian palaeoenvironments and climates. We selected the Ria Zhong (RZ)
section in the Nangqian Basin for palynological analyses, and correlated this section with previous studies from
this and other TP basins. These new results better constrain the biostratigraphy of Paleogene successions across
the plateau, and provide new information on the depositional environment, and elevational and climatic changes
in eastern Tibet during the Eocene. We further synthesise results previously published in Chinese journals,
making these results accessible for an international audience.

**2.   Geological background, stratigraphy and lithofacies**

The Nangqian Basin is located on the border between the Qinghai Province and Tibet Autonomous Region

at an elevation of approximately 4500–5000 m and characterized by a continental seasonal monsoon climate,
with long, cold winters, and short, rainy, and cool to warm summers (Yuan et al., 2017). Most of the annual
precipitation occurs from June to September, when on average, most days in each month experience some
rainfall (Qinghai BGMR, 1991). The region presently hosts alpine steppe and meadow (Fig. 1) characterised by
Cyperaceae, Asteraceae, Amaranthaceae, and Poaceae, as well as conifer and broad-leaved forests dominated by
conifers such as *Pinus*, *Picea*, *Abies*, *Tsuga*, and deciduous angiosperms such as *Quercus* (oak) and *Betula*
(birch) although intensive logging has markedly contracted these forests to steep slopes and remote areas
(Herzschuh, 2007; Baumann et al., 2009).

Although the timing of the Indo-Asian collision remains uncertain (e.g., Xia et al., 2011; Zhang et al.,

2012; Wang et al., 2014), its initiation formed north-eastward extrusion facilitated by motion along a series of
contraction deformation and strike-slip faults in eastern Tibet, including the Yushu–Nangqian thrust belt and the
Jinshajiang strike-slip fault system (Fig. 1; Hou et al., 2003; Yin and Harrison, 2000; Spurlin et al., 2005). The
Nangqian Basin is one of four sedimentary basins in the Nangqian-Yushu region that formed during Paleogene
contraction (Horton et al., 2002), ~80 km-long in S–N direction, and 15 km-wide in E–W direction, and situated
in the eastern part of the Qiangtang terrane (Fig. 1; Hou et al., 2003). The tectonic evolutionary history of the
area includes an early stage extrusion thrust foreland basin, a middle stage strike-slip foreland basin, and the late
stage extrusion strike-slip foreland basin (Wang et al., 2001, 2002; Mao et al., 2010; Jiang et al., 2011).

Paleozoic, Mesozoic, and Paleogene sedimentary rocks exposed along the Yushu-Nangqian traverse

include Carboniferous–Triassic marine carbonates and minor clastic units overlain by Jurassic, Cretaceous, and
Paleogene red beds (Liu, 1988; Qinghai BGMR, 1991). The southern area mainly comprises the Carboniferous
Zhaduo Group ($C^1zd$), whereas the northern area is dominated by younger strata comprising the Upper Triassic
Jieza Group ($T^3jz$; Qinghai BGMR, 1991). Our study concentrated on the Cenozoic gypsum-bearing Gongjue
Formation, which unconformably overlies Carboniferous–Triassic rocks and may be conformable with
underlying Upper Cretaceous strata (Qinghai BGMR, 1983a, 1983b, 1991). It is divided into five lithological
units ($Eg^1$–$Eg^5$), from bottom to top (Du et al., 2011). $Eg^1$ comprises shallow lacustrine facies reaching a
thickness of ca. 400 m, which lie unconformably on a basement of Carboniferous–Permian sedimentary rocks.
The strata in units $Eg^2$, $Eg^4$, and $Eg^5$ were mainly formed in an alluvial environment with rapid sedimentation
rates, with strata reaching a thickness of ca. 530 m, 1100 m, and 2500 m respectively.

The focus of this study is the $Eg^3$ unit which has a more complex depositional history; it is the thickest

(reaching 3500 m) of the five units, and the most widely distributed unit in the Nangqian Basin. $Eg^3$ is divided
into three members: 1) the Ri'Anongguo conglomerate member, which reaches a thickness of approx. 1300 m;
2) the Dong Y'ru sandstone member with limestone beds, which reaches a thickness of 700–1000 m; and 3) the
uppermost Gouriwa member, comprising mudstones (generally developed as red beds) intercalated with gypsum
and reaching 900–1200 m in thickness (Wang et al., 2002). This latter member has been interpreted as being
deposited in a fluviolacustrine environment under a range of climatic conditions (Wang et al., 2001, 2002; Jiang
et al., 2011). Note that the stratigraphic framework for the Gongjue Formation described in Yuan et al. (2017) is
incorrect, referring to the three members above as comprising the entire Gongjue Formation instead of only the
$Eg^3$ unit within the Gongjue Formation; we correct this here based on the descriptions of Wang et al. (2001,
2002) and Du et al. (2011). Based on palynological analyses and ostracod assemblages, the mudstone-dominated
successions of the $Eg^3$ unit have been dated as late Eocene to Oligocene in age (Wei, 1985; Yuan et al., 2017),
which is corroborated by 38–37 Ma $^{40}Ar/^{39}Ar$ ages from interbedded volcanic rocks in the uppermost strata of
the Nangqian Basin (Spurlin et al., 2005).

Though few palynological data currently exist from the Nangqian Basin (Wei, 1985; Yuan et al., 2017),

palynology has been extensively applied for biostratigraphic purposes, as well as to infer Cenozoic climatic
changes, in basins across the TP, including the Qaidam Basin (Xu et al., 1958; Zhu et al.,1985; Wang et al.,
1999; Sun et al., 2005; Lu et al., 2010; Ji et al., 2011; Miao et al., 2011, 2012, 2013a; Cai et al., 2012; Herb et
al., 2015; Wei et al., 2015), Xining Basin (Dupont-Nivet et al., 2008; Miao, 2010; Hoorn et al., 2012; Miao et
al., 2013b; Bosboom et al., 2014), Hoh Xil Basin (Liu et al., 2003; Miao et al., 2016), Tarim Basin (Sun et al.,
1999; Zhu et al., 2005; Bosboom et al., 2011; Wang et al., 2013), Jianchuan Basin (Li L. et al., 2019), and the
Xigaze region of Tibet (Li et al., 2008). Most of these studies are limited to the sedimentary successions within
the foreland basins of the northern TP, rendering it important to gather further data on central Tibetan basins that
preserve a complex sequence of Cenozoic deformation in relation to the Indo-Asian collision zone (Spurlin et
al., 2005). Furthermore, correlation of the above-mentioned northern successions with our new section from the
Nangqian Basin (presented in Section 5.1) is valuable for advancing understanding of differences in vegetational
composition across the TP, as well as the paleoenvironmental and climatic signals recorded by these ecosystems.

**3.  Materials and Methods**

In this study, the RZ section located in the northwestern part of the Nangqian Town (N32°12'10",

E96°27'19.42", altitude 3681 m) was sampled for sedimentological and palynological analyses (Fig. 2). The RZ
section is a ca. 260 m thick portion of the Gongjue Formation where it represents the uppermost Gouriwa
member of the $Eg^3$ unit. The sediments mainly comprise lacustrine facies represented by red mudstones and
siltstones, intercalated with gypsum beds. A more detailed description of the sedimentology, geochemistry, and
palynofacies of the section are presented in a separate manuscript (Yuan et al., submitted). A total of 71
palynological samples were collected from mudstones or fine-grained siltstones.

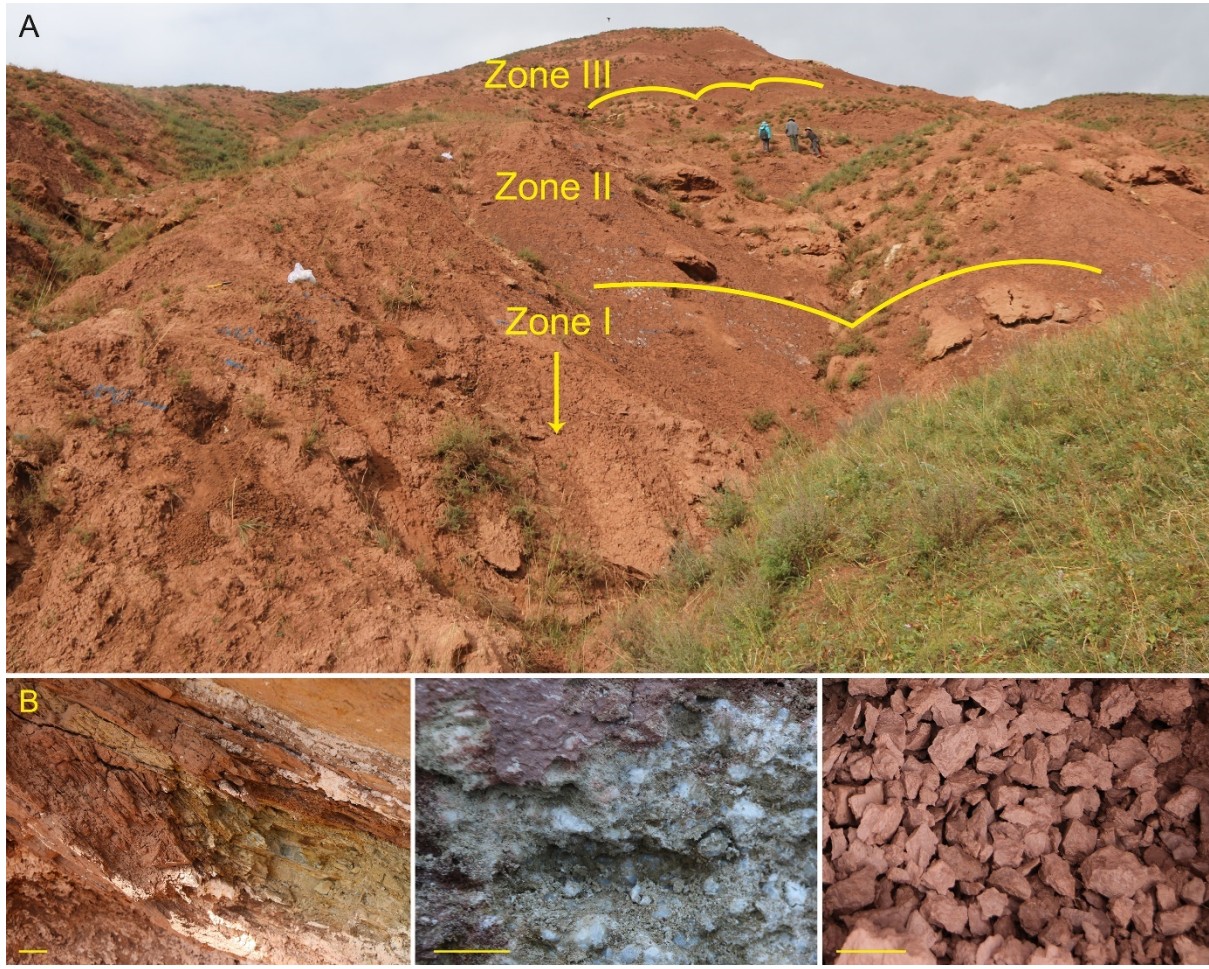

**Figure 2A. Field photograph of the newly sampled Ria Zhong (RZ) section with palynological zones**
**marked. The section is located in the Nangqian Basin, Yushu area, Tibet (N32° 12' 10", E 96° 27' 19.42",**
**altitude 3681 m), and represents the uppermost Gouriwa member of the Eg³ unit in the Cenozoic Gongjue**
**Formation. The lithostratigraphy and geochemistry of the RZ section are further described in Yuan et al.**
**(submitted). B. Field photographs showing representative lithologies in the RZ section, from left:**
**intercalated siltstone and mudstone; massive gypsum; silty mudstone. Scale bar – 5 cm.**

**Table 1. Lithologies of the palynological zones identified within the Ria Zhong (RZ) section in the**
**Nangqian Basin, Yushu area, Tibet. Information is given on the stratigraphic horizons of each zone, its**
**dominant lithologies, and the total number of productive palynological samples that were obtained.**

| Zone | Depth (m) | Dominant lithologies | Productive samples |
|------|-----------|----------------------|--------------------|
| III | 101–16 | Silty mudstones and siltstones with thin gypsum beds | 2 |
| II | 147–101 | Mudstones, nodular mudstones and silty mudstones with thin gypsum beds | 2 |
| I | 251–147 | Mudstones and gypsum | 17 |

The samples were first treated with 36% HCl and 39% HF to remove carbonates and silicates and then
sieved through a 10 µm nylon mesh. Subsequently, the residue was density separated using ZnCl2 (density =
2.1). The organic residue was mounted on microscopic slides in glycerin jelly. All slides were examined at the
Swedish Museum of Natural History under a Leica light-microscope (OLYMPUS BX51), and micrographs were
taken of selected specimens. As is standard for palynostratigraphic studies, we used primarily light microscopy
(LM) to identify, count, and photograph palynomorphs present in the samples. An ESEM FEI Quanta FEG 650
scanning electron microscope (SEM) was used to obtain additional detailed surface images of *Ephedripites*
*(Ephedripites), Ephedripites (Distachyapites),* and other key species. Slides and residues are hosted at the
Swedish Museum of Natural History, Stockholm, Sweden.
From each of the 21 productive samples > 200 grains were identified and counted, and the pollen diagrams
(Fig. 3, Fig. S1 and Fig. S2) plotted using TGView© and Tilia© 2.0 software (Grimm, 1991). We assigned fossil
pollen taxa to Ecological groups or Plant Functional Types (PFTs) according to their correspondence with
nearest living relatives (NLR) in modern Asian biomes (following the taxon-to-NLR assignments of Hoorn et
al., 2012). PFTs are shown in the supplementary dataset of Yuan & Barbolini (2020) as well as Fig. S1 and S2.
Statistical analysis of the palynological assemblages was conducted using CONISS (Constrained Incremental
Sums of Squares cluster analysis), a multivariate agglomerative method for defining zones hierarchically
(Grimm, 1987). A stratigraphically constrained analysis was performed on pollen-percentage values with square
root transformation (Edwards and Cavalli Sforza's chord distance) which up-weights rare variables relative to
abundant ones, and is therefore particularly appropriate for pollen datasets (Grimm, 1987). Results of the
CONISS ordination on all taxa were presented as a dendrogram onto the pollen diagram (Fig. S1), and the
ordination was then repeated to test the robustness of the stratigraphic zones by excluding the "Other / Unknown
/ Unresolved NLR" ecological group. Very similar zones were retained in the new cluster analysis (Fig. S2),
increasing confidence that these zones represent true changes in vegetation and climate dynamics recorded
throughout the section. Both CONISS ordinations were used in conjunction with the taxonomic and quantitative
composition of the palynological assemblage, in order to demarcate zones and subzones within the section.

## 4. Results

Recovery of palynomorphs was generally poor, particularly in the upper part (0–147 m) of the section.
Although there is no direct linkage between the productivity of pollen samples and lithofacies, evidence of
increasing aridity is preserved in the upper part of the section through both the palynological and
sedimentological records. Accordingly, vegetation biomass was likely lower in these more arid environments,
and extended exposure on the landscape before burial can destroy palynomorphs; both of these factors are likely
contributors to the lower productivity observed. In total, only 21 productive samples were obtained from 71
processed samples, indicating a productivity ratio of 30%. Nevertheless, individual well-preserved palynological
assemblages were recovered throughout the section, enabling a representative portrayal of vegetation changes
through time to be reconstructed. In total 26 spore and 81 pollen taxa (5 gymnosperm and 76 angiosperm
morphospecies) were able to be identified, which are illustrated (Plate I, II, III) and grouped into seven different
Plant Functional Types (PFTs) that represent various ecological groups (Fig. 3). Overall trends for the RZ
section include rare conifers and a general dominance of steppe-desert pollen in all zones. Ferns are abundant
and diverse, particularly in the lower part of the section (Zone I), while temperate and warm broad-leaved forest
are relatively diverse and present throughout, but not particularly abundant in any zone. Steppe-desert pollen
decreases concurrently with a spike in tropical forest pollen in one sample from Zone II, and then resurges to
dominance in Zone III. Palynological zones are marked on the field photograph of the RZ section (Fig. 2) and
plotted on the pollen diagrams (Fig. 3, Fig. S1 and Fig. S2).
While the generally high proportion of spores suggests a significant proportion of local deposition (at site),
as a whole the palynological assemblages are taken to reflect the regional vegetation, and may also include some
taxa that are prone to longer-distance transport. These latter taxa are mostly trees, and are normally present in
small percentages except for *Pinus*, which can comprise 10–50% in the palynological records of deserts and
steppe–deserts (but is extremely rare in our section; Ma et al., 2008; Hoorn et al., 2012). Studies on the
correspondence between the modern pollen rain and regional vegetation on the Tibetan Plateau indicate
generally good agreement, and confirm that the use of palynology for palaeoenvironmental reconstruction in
deep time is therefore also appropriate (Cour et al., 1999; Li et al., 2020).

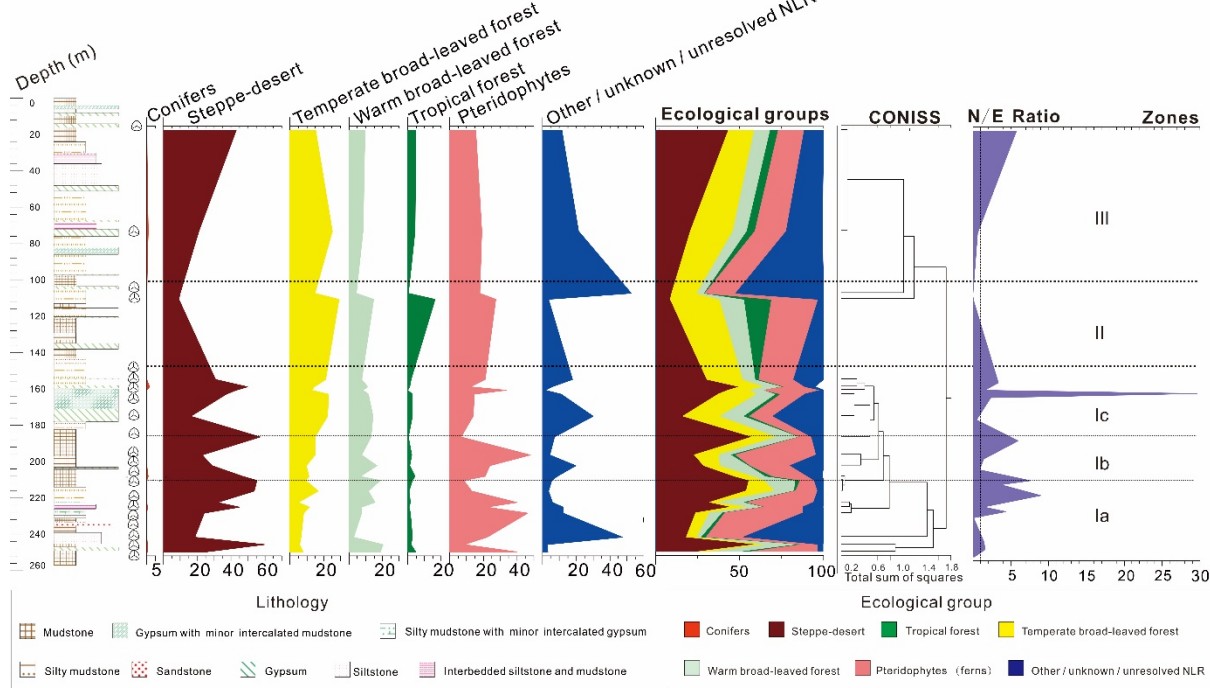

Figure 3: Cumulative pollen summary diagram of the Ria Zhong (RZ) section in the Nangqian Basin, Yushu area, Tibet, with palynomorph percentages of the total pollen sum plotted on the x-axis, and zones and subzones based on CONISS ordinations. Pollen taxa are grouped in Plant Functional Types (PFTs) according to their correspondence with nearest living relatives (NLR), indicated in the legend. Some taxa have multiple or unresolved botanical affinities, and are thus assigned to the "Other / unknown / unresolved NLR" group. Productive horizons are indicated by a small trilete spore to the right of the simplified section log. The *Nitraria/Ephedra* (N/E) pollen ratio is plotted in purple, with a dashed line indicating the transition point between desert/semi-desert ecosystems (< 1) and steppe-desert (> 1).

## 4.1 Stratigraphic zonation based on palynology

Based on results of two CONISS ordinations combined with the taxonomic and quantitative composition of the palynological assemblage (see Methods section; Fig. 3, Fig. S1 and Fig. S2), the succession was divided into three zones (I, II, III) of which Zone I was further divided into three subzones (a, b, c), all of which demonstrate unique vegetation dynamics within that zone. Important trends for each zone and subzone are described below. The zone boundaries are positioned at the upper limit of the samples that mark each boundary. A complete overview of the raw counts, percentages and arithmetic means are given in the supplementary information.

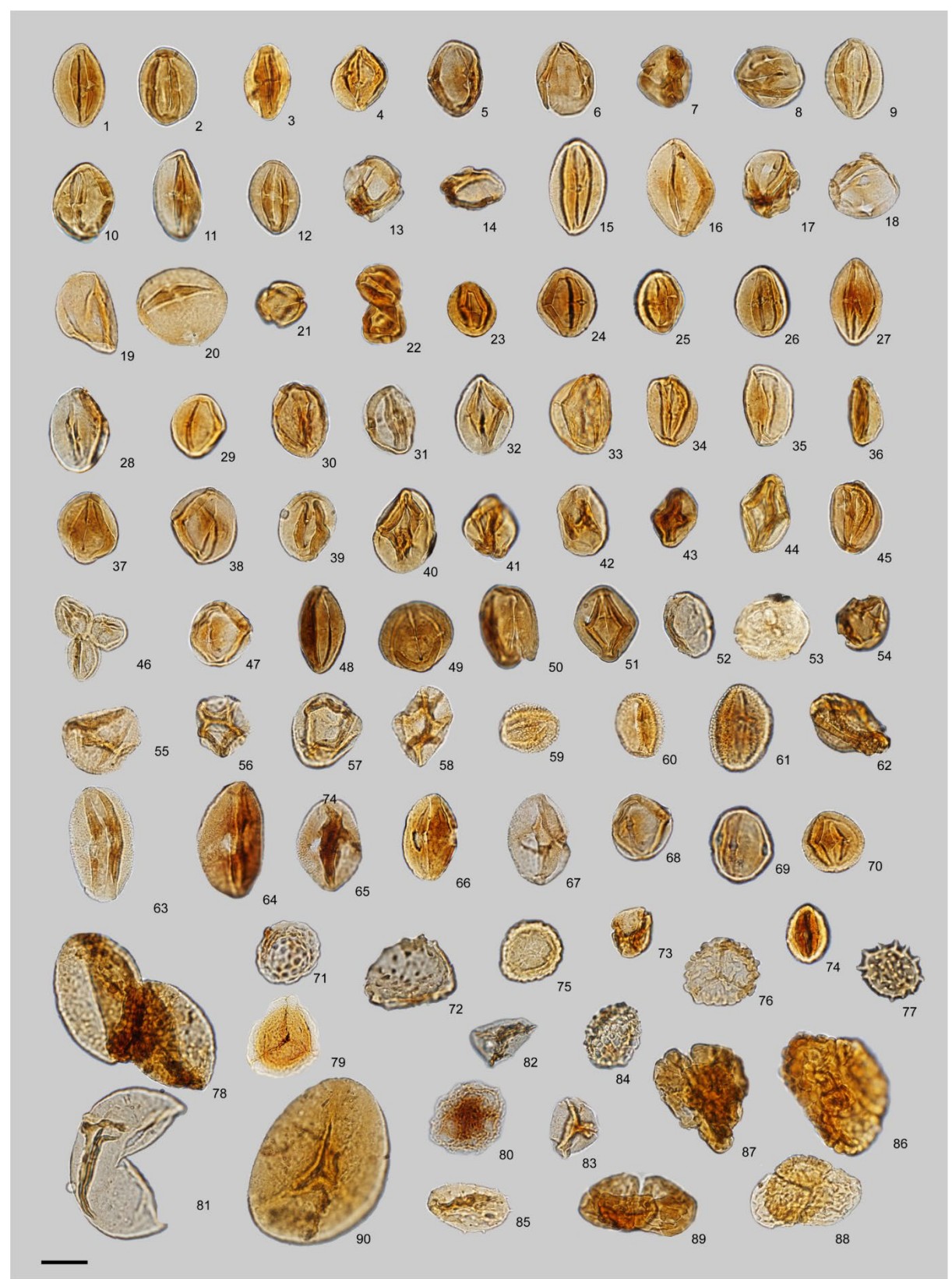

**Plate I: Light micrographs of selected pollen grains and spores from the Ria Zhong (RZ) section, Nangqian Basin.**

**Scale bar – 10µm. 1-12.** *Nitrariadites/Nitraripollis.* **13-20.** *Meliaceoidites.* **21-25.** *Qinghaipollis.* **26-32.** *Rhoipites.* **33-36.**

*Labitricolpites.* **37-45.** *Quercoidites.* **46.** *Quercoidites minutus.* **47-51.** *Rutaceoipollenites.* **52-54.** *Momipites.* **55-58.**

*Fupingopollenites.* **59-61.** *Ilexpollenites.* **62.** *Aceripollenites.* **63-67.** *Euphorbiacites.* **68-69.** *Faguspollenites.* **70.**

*Retitricolporites.* **71.** *Chenopodipollis.* **72.** *Echitriporites* **sp. 73.** *Sporopollis.* **74.** *Caprifoliipites / Oleoidearumpollenites?.*
**75-76.** *Pterisisporites.* **77. Unidentified baculate spore. 78.** *Liliacidites.* **79-80.** *Pterisisporites.* **81.** *Taxodiacites.* **82-83.**
*Deltoidospora.* **84.** *Lycopodiumsporites.* **85.** *Spinizonocolpites.* **86-88.** *Verrucosisporites.* **90.** *Lygodiumsporites.*


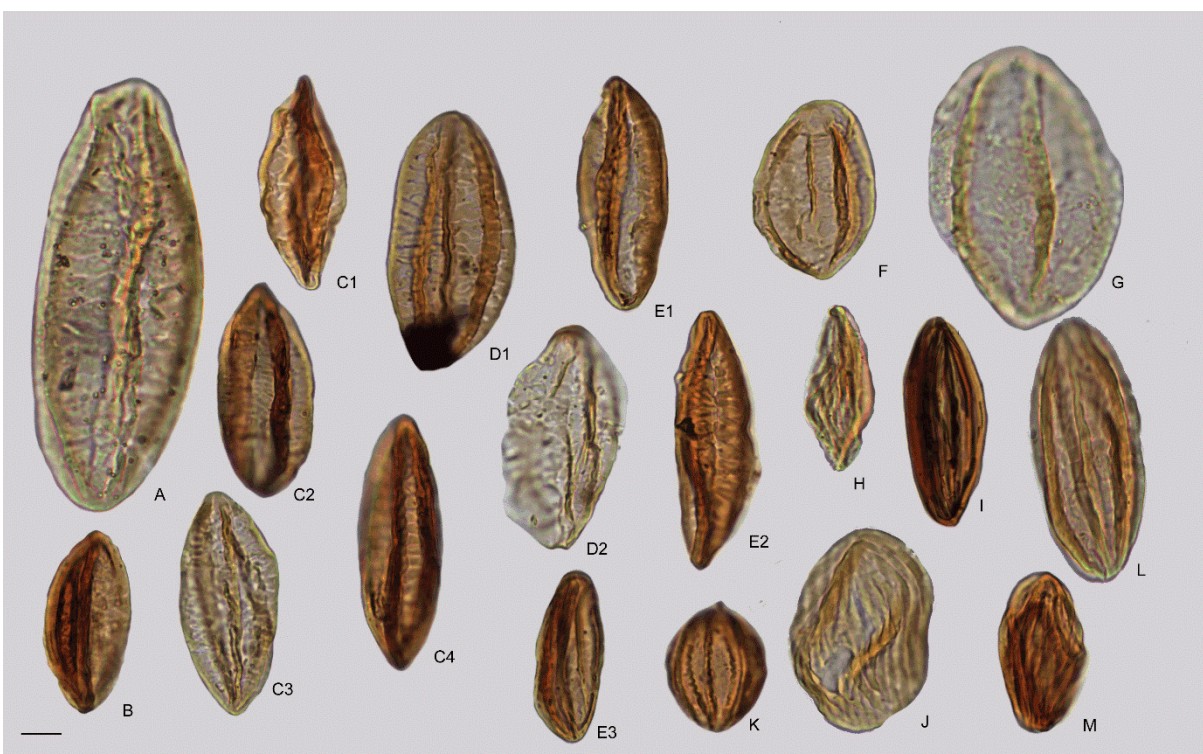


**Plate II: Light micrographs of ephedroid pollen from the Ria Zhong (RZ) section, Nangqian Basin. Scale bar – 10μm.**
**A.** *Ephedripites (Distachyapites) cheganica.* **B.** *Ephedripites (Distachyapites) fusiformis.* **C1-C4.** *Ephedripites*
*(Distachyapites) megafusiformis.* **D1-D2.** *Ephedripites (Distachyapites) eocenipites.* **E1-E3.** *Ephedripites (Distachyapites)*
*nanglingensis.* **F.** *Ephedripites (Distachyapites) obesus.* **G.** *Ephedripites (Ephedripites) bernheidensis.* **I.** *Ephedripites*
*(Ephedripites)* **sp. 2 (Han et al., 2016). K.** *Ephedripites (Ephedripites)* **sp. b. H.** *Ephedripites (Ephedripites)*
*montanaensis.* **J.** *Ephedripites (Ephedripites)* **sp. a. L.** *Steevesipollenites cf S. binodosus.* **M.** *Steevesipollenites*
*jiangxiensis.*

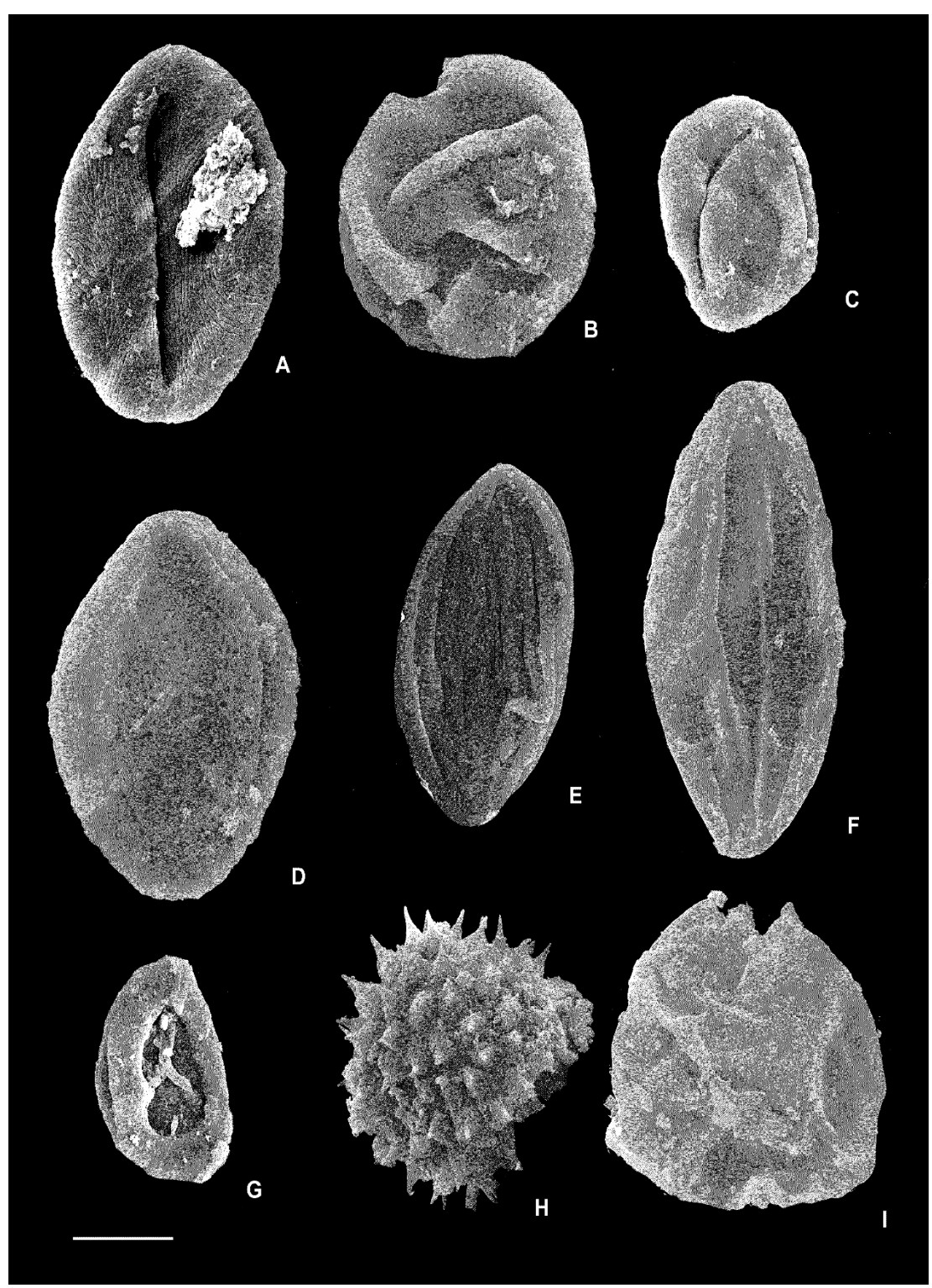


**Plate III: Scanning Electron Microscope (SEM) photographs of selected fossil taxa in the Ria Zhong (RZ) section, Nangqian Basin. Scale bar – 10μm. A, B, C. *Nitrariadites/Nitraripollis*. D. *Retitricolporites*. E. *Ephedripites (Ephedripites)* sp. 2 (Han et al., 2016). F. *Ephedripites (Distachyapites) eocenipites*. G. *Pterisisporites*. H. Unidentified baculate spore. I. *Momipites*.**

**4.1.1 Zone I (17 samples, 251–147 m)**

Conifers in this zone are rare, represented only by *Taxodiacites* (Cupressaceae) and *Tsugaepollenites* (Pinaceae), and never comprising more than 3%. The assemblage is dominated by steppe-desert taxa, which together comprise nearly 40% and include numerous types of *Ephedripites* (Plate II), *Nitrariadites/Nitraripollis*, and *Qinghaipollis*, together with more rare xerophytic taxa such as *Chenopodipollis* and *Nanlingpollis*. The second most abundant group is the Pteridophytes (ferns), which is also the most diverse of all the groups represented in the RZ section. Broad-leaved forest forms a minor component of the palynological record, with warm forest being more abundant than temperate forest and represented primarily by *Rutaceoipollenites*. Tropical forest pollen is rare, and includes *Spinizonocolpites* and *Fupingopollenites*. Some pollen types have unresolved botanical affinities or affinities with multiple ecological groups, and these are grouped separately but do not provide ecological information.

Zone I is divided into three subzones on the basis of abundance patterns among particular palynomorph taxa. Subzone **Ia** (9 samples, 251–209 m) is unique in that *Ephedripites* (steppe-desert group), *Cupuliferoipollenites* (temperate broad-leaved forest), and *Rutaceoipollenites* (warm broad-leaved forest) are more abundant than in other subzones of Zone I, while *Momipites / Engelhardthioipollenites* (warm broad-leaved forest) is less abundant, and *Aceripollenites + Faguspollenites* (temperate broad-leaved forest) are very rare compared to the remainder of Zone I. Of the entire section, *Caryophyllidites* (steppe-desert) only occurs in **Subzone Ib** (3 samples, 209–187 m), which also records a spike of *Momipites/Engelhardthioipollenites* (warm broad-leaved forest). **Subzone Ic** (5 samples, 187–147 m) contains the greatest proportion of *Nanlingpollis* (steppe-desert) in the entire section, as well as spikes of *Aceripollenites + Fraxinoipollenites* (temperate broad-leaved forest), while *Qinghaipollis* (steppe-desert) and ferns decrease in this subzone.

**4.1.2 Zone II (2 samples, 147–101 m)**

No conifer pollen occurs in this zone, and on average, the steppe-desert taxa *Ephedripites* (gymnosperm)*, Nitrariadites/Nitraripollis* and *Qinghapollis* (angiosperms) are far less abundant than in other parts of the section (average 9% in Zone II vs 38% (Zone I) and 32 % (Zone III)). However, a spike in the ancestral (old) *Ephedra* type is observed during Zone II, which is not observed in the other zones or later in the Eocene (Yuan et al., 2017). Notably, tropical forest pollen increases markedly in one sample from this zone (as regional input), comprising mostly *Fupingopollenites*, while temperate broad-leaved forest (*Aceripollenites*, cf. *Caprifoliipites*) and warm broad-leaved forest (*Rutaceoipollenites*) are also more prevalent. Pollen of unknown or multiple

affinities is higher in this zone, and reflected by spikes of *Labitricolpites* and *Rhoipites*. A low recovery of productive samples was obtained from this zone, and the above-described trends may thus reflect an incomplete picture of environmental changes during this interval.

**4.1.3 Zone III (2 samples, 101–16 m)**

Conifers in this zone are very rare, represented only by *Tsugaepollenites*. Steppe-desert taxa again dominate this zone, with *Nitrariadites/Nitraripollis* increasing steadily through the section. Temperate broad-leaved forest is now much more common than warm broad-leaf or tropical forest pollen, while ferns are least common in this zone but still plentiful. A low recovery of productive samples was obtained from this zone, and the above-described trends may thus reflect an incomplete picture of environmental changes during this interval.

## 5. Discussion

### 5.1 Age assignment

Age constraints for the RZ section are provided by the K–Ar ages from shoshonitic lavas and felsic and porphyry intrusions that are either interbedded with, or unconformably overlie, the lacustrine to alluvial Nangqian strata. Emplacement ages across the Nangqian Basin vary between 32.04–36.5 Ma (Deng et al., 1999); $37.0 \pm 0.2$ Ma–$38.2 \pm 0.1$ Ma (Spurlin et al., 2005); 37.1–37.8 Ma (Zhu et al., 2006); and $35.6 \pm 0.3$–$39.5 \pm 0.3$ Ma (Xu et al., 2016). In the latter study, zircon U–Pb age data were derived from felsic intrusions sampled at two localities in the Nangqian Basin (Boza and Nangqian). The syenite porphyries from the Boza area (further south of the RZ section) show an emplacement age of $35.58 \pm 0.33$ Ma, while the monzonite porphyries from the Nangqian area (just southeast of the RZ section) have older magmatic emplacement ages, ranging from $39.5 \pm 0.3$ Ma to $37.4 \pm 0.3$ Ma. As this age range is broadly coeval with the age of the mafic volcanic rocks in the Nangqian Basin (37.0–38.2 Ma; Spurlin et al., 2005) as well as the age range obtained by Zhu et al. (2006), here we consider ~37–38 Ma to represent a minimum age for the RZ section. This is also congruent with palynological evidence for the overall age of the sampled strata (Fig. 4), which is discussed in more detail below.

The assemblage from the RZ section is very similar to those from the Yang Ala section in the Nangqian Basin, dated as late Eocene (Yuan et al., 2017), the Eocene Wuqia assemblage (site 98) from the west Tarim Basin (Wang et al., 1990a; 1990b), the late middle Eocene–late Eocene assemblage from the upper Niubao

Formation, Lunpola Basin (Song and Liu, 1982; Li J.G. et al., 2019), and the Bartonian (41.2–37.8 Ma) part of
the palynological record in the Xining Basin (Dupont-Nivet et al., 2008; Hoorn et al., 2012; Han et al., 2016).
Specifically, the absence of *Classopollis*, *Exesipollenites*, and *Cycadopites* combined with the predominance of
*Nitrariadites/Nitraripollis* and *Ephedripites* pollen, and the presence of the middle Eocene–Neogene genus
*Fupinggopollenites* (Liu, 1985), indicates that the RZ section cannot be older than middle Eocene (Fig. 4). It is
also unlikely to be of latest Eocene age or younger due to the lack of significant conifers that become more
common approaching the Eocene–Oligocene Transition (Hoorn et al., 2012; Page et al., 2019; Fig. 4). Specific
ranges and abundance patterns of these and other key taxa within Eocene Tibetan basins (Fig. 4; Fig. 5) enable
the age of the section to be better constrained, which is explored in greater detail below.

*Ephedra* is a gymnosperm shrub with the oldest macrofossils from the Early Cretaceous (Bolinder et al.,

2016; Han et al., 2016) but the genus is probably older, dating to the Triassic (Yang, 2002; Sun and Wang, 2005)
or even the Permian (Wang, 2004) based on the ephedroid pollen record. Its current distribution is limited
primarily to arid and semiarid regions of the world (Stanley et al., 2001), and the fossil pollen representative,
*Ephedripites*, is widespread in Cenozoic evaporates, indicating the xerophytic nature of this genus (Sun and
Wang, 2005). The Xining Basin in northern Tibet records a particularly time-extensive section with good age
control (Dupont-Nivet et al., 2008, 2008; Hoorn et al., 2012; Meijer et al., 2019) that reveals a detailed pattern
of changes in *Ephedripites* pollen during the middle–late Eocene. After 38.8 Ma, *Ephedripites* comprised ca.
20–60% of the total palynological composition in the Xining Basin, with a predominance of the derived type,
*Ephedripites* subgen. *Distachyapites* (Han et al., 2016). Prior to this (ca. 41–38.8 Ma), the record comprised a
mix of the derived type, the ancestral type (*Ephedripites* subgen. *Ephedripites*), and another ephedroid genus,
*Steevesipollenites* (Han et al., 2016; Bolinder et al., 2016). A similar pattern is observed in the Nangqian Basin,
with a spike of the ancestral type of *Ephedra* only recorded in Zone II, and not observed in the rest of the RZ
section or elsewhere in the Nangqian Basin (Yuan et al., 2017). This suggests a correlation between Zone I of
the Xining Basin with Zone II of the RZ section (Fig. 5). As it is possible that the change in *Ephedripites*
diversity may not have occurred across Tibet simultaneously (i.e., at ~39 Ma), we suggest that this most likely
constrains the age of the RZ section to late Eocene (Bartonian; 41.2–37.8 Ma).

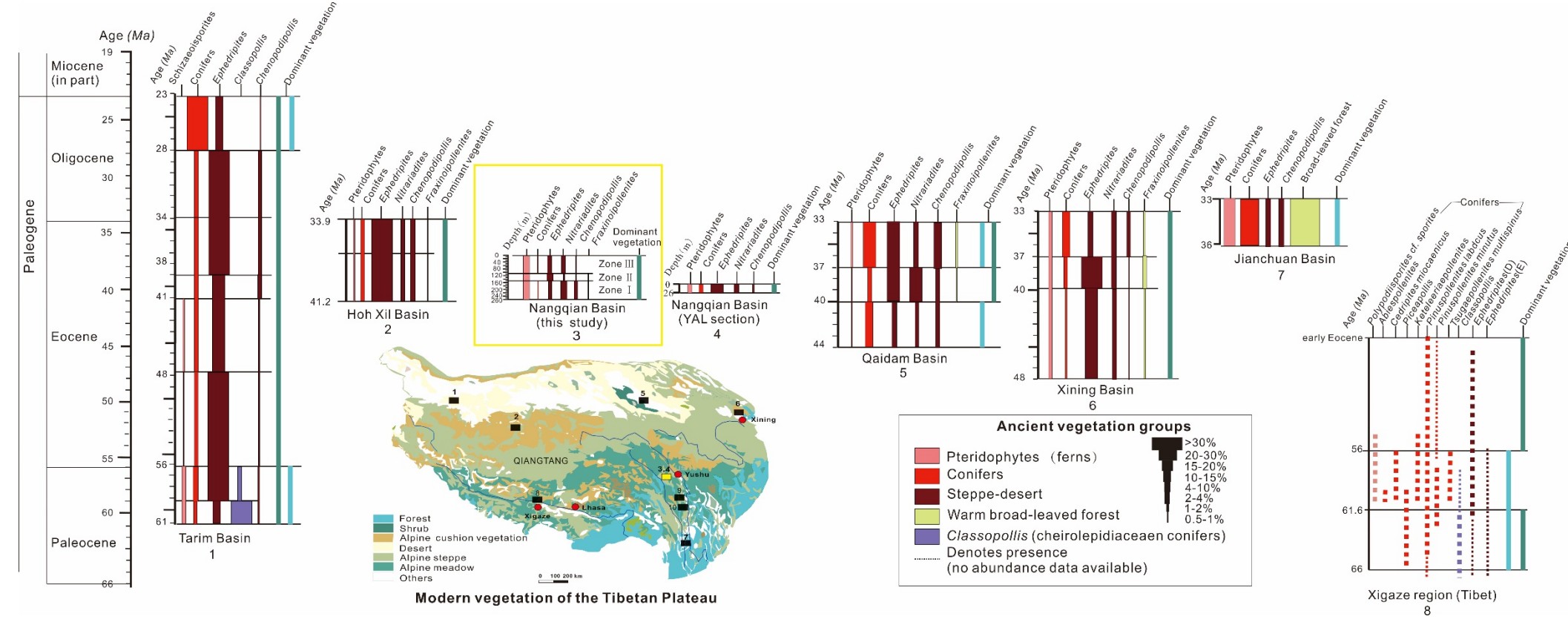


Figure 4: Palynozonation of the Paleogene successions across the northern, central, and southern TP, with numbers under each section indicating the associated basin: 1. Tarim Basin (Wang et al., 1990a; 1990b); 2. Hoh Xil Basin (Miao et al., 2016); 3, 4. Nangqian Basin (this study; Yuan et al., 2017). 5. Qaidam Basin (Lu et al., 1985; Zhang et al., 2006; Miao et al., 2016); 6. Xining Basin (Wang et al., 1990a; 1990b; Hoorn et al., 2012); 7. Jianchuan Basin (Wu et al., 2018); 8. Xigaze Basin (Li et al., 2008). The dominant ancient vegetation reconstructed from palynological assemblages is shown to the right of each section. Modern vegetation map redrawn from Baumann et al. (2009).

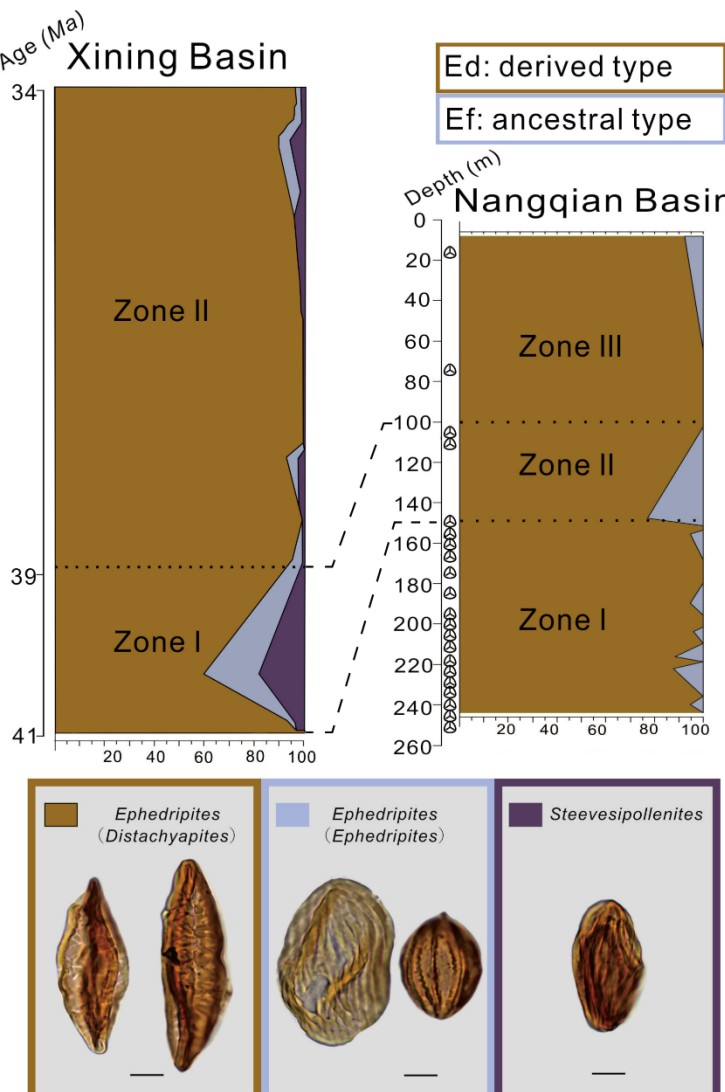

**Figure 5: Eocene ephedroid pollen composition in the Xining (northeastern TP) and Nangqian (east-central TP)**
**basins, illustrating the distributions of *Ephedripites* subgen. *Ephedripites* (ancestral type; "Ef"), *Ephedripites* subgen.**
***Distachyapites* (derived type; "Ed"), and *Steevesipollenites*. Productive horizons for the Ria Zhong (RZ) section are**
**indicated by a small trilete spore to the right of the marked depths.**

In addition to the proportions of the ancestral vs. derived type of *Ephedripites*, a significant spike in

tropical forest pollen in one sample at this time, combined with a large decrease in steppe-desert pollen, suggests

that Zone II of the RZ section reflects a temporary warming interval in the Eocene. Although the increase in

tropical forest taxa in in one sample from this zone does not indicate an actual biome shift in the Nangqian

region from "steppe" to "tropical forest", it suggests a change in regional climate through increased input of

regional tropical taxa. This could possibly be concurrent with the MECO (~40 Ma), a transient warming event

that preceded rapid aridification in Central Asia (driven primarily by proto-Paratethys sea retreat; Kaya et al.,
2019). This interval is followed by a change in lithofacies (decreasing thickness of gypsum beds) and an
increase in steppe-desert pollen records in northwestern China (Bosboom et al., 2014). Similar trends are also
observed in the Nangqian Basin (Fig. 3), suggesting a possible correlation. However, it must be considered that
the upper zones of the RZ section yielded a low number of samples (Zone II and III each comprise only 2
samples), and the tropical forest spike is only present in one of these samples. This places statistical limitations
on the interpretations that can be drawn, therefore further investigations should be made in Nangqian and other
parts of Tibet to corroborate this finding. Accordingly, for the moment we do not date the RZ section on the
basis of a tentative correlation to the MECO at ~40 Ma; however, available evidence does suggest that the spike
of tropical forest represents a temporary shift in regional climate. The palynomorphs from these samples were
not degraded or compressed to a greater degree than palynomorphs from the rest of the section, and of a similar
colour and appearance, suggesting it is unlikely that the pollen in Zone II represents reworking or
contamination. Furthermore, the increase in tropical forest taxa is accompanied by a large decrease in steppe-
desert pollen which is not observed in the other zones of this section (average 9% steppe-desert pollen in Zone II
vs 38% (Zone I) and 32 % (Zone III)), nor later in the Eocene in the Nangqian Basin (Yuan et al., 2017). This
further indicates a temporary shift in the regional climate to warmer and wetter at this time.

In northern Tibet, Pinaceae (conifers) abruptly increased in the palynological record at 36.55 Ma (Page et

al., 2019), which is not observed in the RZ section. The rare conifers in this latter assemblage are in accordance
with the minimum depositional age constraints of ~37–38 Ma from overlying volcanic rocks. In conjunction
with the palynostratigraphic correlations from across Tibet (Fig. 4), as well as the change in the proportions of
the ancestral vs. derived type of *Ephedripites* (Fig. 5), the age of the complete section is proposed to be
Bartonian (41.2–37.8 Ma; Fig. 4; Fig. 5).

**5.2 Paleoclimate**

The RZ section records three distinct palaeofloras in east-central Tibet that evolved in response to changing

climate in the Eocene (Fig. 6). During deposition of Zone I, the climate was warm, and vegetation was
characterised by steppe-desert shrubs, diverse ferns, and a lesser component of temperate and warm broad-
leaved forest. Interestingly, prominent vegetation groups with very different moisture requirements existed
within a limited distance of each another in the Nangqian area. A very diverse and abundant pteridophyte (fern)
community, as well as conifers such as *Taxodiacites* and *Tsugaepollenites* would have required higher humidity
(Liu et al., 2012; Kotthoff et al., 2014), but the abundant halophytic and xerophytic steppe-desert vegetation
would likely only have been competitive in arid environments. The dominant plants belonging to these salt- and
drought- tolerant groups (*Nitraria* and *Ephedra*) grow today in Central Asian regions with MAP of 100mm or
less, and are also associated with arid palaeoenvironments through the Cenozoic (Sun and Wang, 2005).
Although the conifers (produced by cypress and *Tsuga*) could have been windblown from further distances, the
coexistence of such diverse and abundant ferns and steppe-desert vegetation in the landscape, PFTs with
opposing moisture requirements for competitiveness, has not been observed in other Tibetan basins to date
(Miao et al., 2016, Table 1), and therefore seems not to reflect conventional spatial patterning of less water-
dependant vegetation growing upland. Rather, it may suggest an environment with strongly seasonal
precipitation that would favour lush vegetation growth for a restricted interval and alternately, xerophytic
vegetation during the dry season.

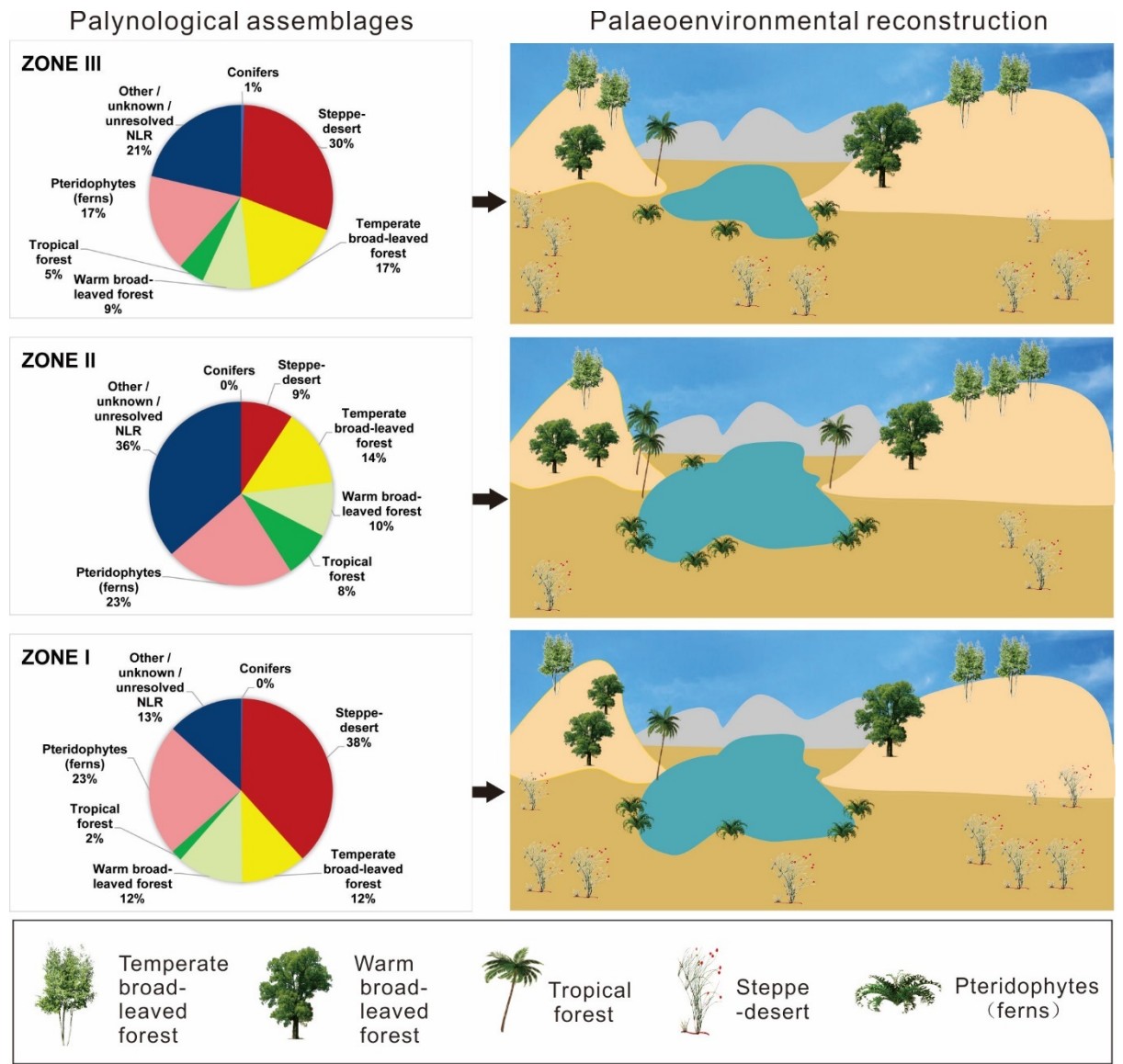

**Figure 6: Palaeoenvironmental reconstruction of the Nangqian area, illustrating the three distinct floral assemblages recovered from the RZ section. Vegetation during deposition of Zone I was dominated by steppe-desert shrubs, which decreased sharply in Zone II in conjunction with a spike in tropical forest. Afterwards the basin became drier and steppe-desert vegetation again dominated the landscape.**

Based on a comparison of existing palynofloral records with our new section, the northern regions of the plateau (Tarim, Qaidam, Hoh Xil, and Xining basins) were already significantly more arid than the central TP in the middle Eocene, having hosted greater proportions of xerophytic plants (Fig. 4). Therefore, precipitation in the greater Nangqian region would have been unlikely to derive from the westerlies, which served as the dominant moisture source northward of the central TP since at least the early Eocene (Caves et al., 2015). This suggests that the central TP could have instead been influenced by a southern monsoon system similar to the modern I-AM in the middle–late Eocene, although not to the degree experienced by southern Tibet, which

hosted greater proportions of forest and was likely more humid (e.g., Jianchuan Basin; Fig. 4). However, it
should be borne in mind that rainfall seasonality is not always a proxy for the existence of monsoons; although
leaf form is the preferred method for detecting monsoons in deep time climates (Spicer et al., 2017), the absence
of well-preserved fossil leaf assemblages from the Nangqian Basin to date prevents this comparison.
Furthermore, palynological records alone are not sufficient for detecting whether the nature of monsoons in the
Eocene was more similar to the present I-AM or South Asian Monsoon (SAM), which contributes mostly to the
moisture in the Nangqian region today (Li L. et al., 2019).

Our results indicate that the brief warming interval recorded in Zone II prompted a considerable change in

the vegetation in east-central Tibet, encouraging the temporary spread of (dry) forests in the region, while
steppe-desert vegetation contracted. Warming is reflected by an atypical spike in tropical forest, while a warm
broad-leaved forest spike in northeastern Tibet is coincident with the MECO (Hoorn et al., 2012; tropical forest
is exceedingly rare in the latter area during the middle–late Eocene). In order to estimate relative humidity in
arid environments such as these, the *Nitraria/Ephedra* (N/E) ratio can be used to distinguish between
desert/semi-desert (< 1) and steppe-desert (> 1; Li et al., 2005; Hoorn et al., 2012). Although both genera
occupy arid environments today, *Ephedra* is currently distributed primarily throughout deserts, semi-deserts and
grasslands globally (Stanley et al., 2001), while *Nitraria* is a relatively more humid steppe-desert taxon (Cour et
al., 1999; Sun and Wang, 2005; Jiang and Ding, 2008; Li et al., 2009; Zhao and Herzschuh, 2009).

In the RZ section, the proportion of temperate broad-leaved forest in relation to warm broad-leaf and

tropical forest became much greater in the upper part (Fig. 3), indicating a cooler climate in the late Eocene,
which matches cooling trends recorded by clumped isotopes both in the Nangqian Basin (Li L. et al., 2019) and
in the Xining Basin (Page et al., 2019). Importantly, the N/E ratio in the RZ section is lowest immediately
following the warming interval in Zone II (Fig. 3) and persists for an extended period, indicating rapid,
prolonged aridification. An overall expansion of steppe-desert vegetation is observed in Zone III, corresponding
with patterns observed on the northeastern TP in the late Eocene (Hoorn et al., 2012; Bosboom et al., 2014).
Accordingly, our vegetation results have implications for understanding the importance and extent of
aridification across Central Asia in the late Eocene, which was primarily driven by proto-Paratethys Sea
regression (Kaya et al., 2019). Ecosystem responses to this event on both the northeastern and east-central parts
of the TP demonstrates that aridification across the Asian continental interior in the late Eocene could have been
further-reaching than previously thought. Our findings show that after sea regression, westerly moisture supply
carried from the proto-Paratethys Sea was reduced as far as central Tibet. This provides further support for the

argument that this sea was a major source of moisture for the Asian interior, and thus a primary driver of Central Asian climate during the Eocene (Bosboom et al., 2014; Bougeois et al., 2018; Kaya et al., 2019; Meijer et al., 2019).

Long-term aridification in the late Eocene exerted further influence on vegetational composition in east-central Tibet with regards to the proportions of the ancestral vs. derived types of *Ephedripites*. In modern and Quaternary settings, this has been developed as a ratio to distinguish between desert and steppe-desert environments, termed the *Ephedra fragilis*-type s.l./*Ephedra distachya*-type (Ef/Ed) ratio (whereby *E. fragilis* represents the ancestral type and *E. distachya*, the derived type; Fig. 5). Tarasov et al. (1998) found the *E. fragilis*-type s.l. to be common in arid climates with mean temperatures of the warmest month above 22°C. Herzschuh et al. (2004) applied the Ef/Ed ratio to Holocene pollen spectra from the Alashan Plateau and tested its reliability with a regional modern pollen dataset, finding Ef/Ed ratios > 10 in most samples from desert sites, and values < 5 in most samples from the sites with more favourable climates (e.g., forest-steppe, steppe, and alpine meadow).

In the middle–late Eocene of Central Asia, the ancestral type of *Ephedripites* never comprises more than 25% of the ephedroid pollen sum in northeastern Tibet while the derived type makes up at least 60% (Xining Basin; Han et al., 2016 and Qaidam Basin; Zhu et al., 1985; Miao et al., 2013a; Jiuquan Basin; Miao et al., 2008), and this also appears true for northwestern Tibet (Tarim Basin; Wang, et al., 1990b; Hoh Xil Basin; Miao et al., 2016) and east-central Tibet (Yuan et al., 2017; this study). Therefore, Ef/Ed ratios > 10 (supposedly indicative of desert ecosystems) are never observed, despite the N/E ratio indicating regular existence of deserts or semi-deserts in northern Tibet (Zhu et al., 1985; Hoorn et al., 2012; Miao et al., 2016), and central Tibet (Yuan et al., 2017; this study) in the Paleogene. Sedimentological evidence suggests the N/E ratio to be more reliable for these deep time environments, with *Nitraria* and *Ephedra* pollen being widely distributed in evaporites and red beds indicating deposition in arid or semi-arid climates (Sun and Wang, 2005). Therefore, while pollen ratios appear to reflect reliable functions of climate and landscape change for modern and Holocene settings (Li et al., 2010), our results identify possible contradictions between the N/E and Ef/Ed pollen ratios. This indicates that further verification of these pollen ratios in modern settings and across larger spatial scales is necessary for reliable palaeoenvironmental reconstructions in deep time.

A comparison of palynological assemblages across the Qinghai-Tibetan region indicates that vegetation has changed markedly from the Paleogene to the present (Fig. 4). While the Nangqian region was dominated by steppe-desert shrubs in the past, it now hosts primarily alpine biomes, as do the Hoh Xil and Xining basins. In

contrast, the Tarim and Qaidam basins are now significantly more arid than in the Eocene, and forest- and shrub-
steppe have been replaced with desert vegetation (Fig. 4). The Jianchuan Basin to the south was dominated by
mixed tropical-subtropical coniferous and broad-leaved forest (Wu et al., 2018), and is also forested today (but
with species of a less thermophilic nature). Similarly, the Markam and Gonjo basins host alpine meadow and
forest today; although detailed palynological records have not yet been recovered, macrobotanical fossils
suggest these areas were dominated by mixed broad-leaved and coniferous forest in the late Eocene–early
Oligocene (Su et al., 2018; Studnicki-Gizbert et al., 2008). The above changes indicate that late Paleogene and
Neogene topographic growth (creating new high-elevation biomes; Fig. 1A and B), the aridification of inner
Asia (Caves et al., 2014, 2016), and global cooling (Zachos et al., 2001; DeConto and Pollard, 2003; Pagani et
al., 2011) were all drivers of Cenozoic vegetation shifts across the TP.

**5.3 Elevational implications**
High-altitude conifers are rare in this particular record, although the high-elevation genus *Tsugaepollenites*
(Fauquette et al., 2006) is present. This could be driven by four possible factors: 1) taphonomy i.e., the
assemblage has a high proportion of autochthonous spores and pollen with little input from the peripheral
mountains, 2) elevation of this region was relatively low in the middle–late Eocene (< 3000m as proposed by
Botsyun et al., 2019; also see Wei et al., 2016), 3) due to the generally wetter climate in relation to the
northeastern plateau basins, conifers are not competitive and surrounding mountains are instead forested by
temperate angiosperms, and 4) central Tibet recorded regional pollen transported by different atmospheric
circulation systems.
Regarding the first possibility, conifers are windblown and can be transported far distances (Lu et al., 2008;
Ma et al., 2008; Zhou et al., 2011); as the region already likely experienced a monsoonal climate (Spicer, 2017;
Licht et al., 2014; Caves et al., 2017; this study) we consider it unlikely that our assemblages record little to no
regional vegetation. The second factor, elevation history of the TP, is a controversial topic of discussion, and
palynological evidence from the RZ section does not provide strong support either for or against a relatively low
middle–late Eocene palaeoaltitude in the region. Although the upper part of the RZ section in the Nangqian
Basin likely just pre-dates the high-elevation signal further to the north from 37 Ma onwards (Dupont-Nivet et
al., 2008; Hoorn et al., 2012; Page et al., 2019), an expanding body of data indicates that a proto-Tibetan
Highland with complex topography was already in place during the Paleogene (Xu et al., 2013; Ding et al.,
2014; Wang et al., 2014; Valdes et al., 2019).
Isotopic evidence suggests moderate to high elevations for the Nangqian Basin in the late Eocene (valley
floor 2.7 (+0.6/–0.4) km above sea level; surrounding mountains 3.0 ± 1.1 km above sea level; Li L. et al.,
2019). In the adjacent Gonjo Basin, stable isotope data suggest the basin had already attained 2100–2500 m
palaeoelevation by the early Eocene (Tang et al., 2017). Some of the broad-leaved angiosperms trees present in
the new Nangqian assemblage could have grown at maximum elevations of 3600–4000 m during the Eocene
(*Ilex, Quercus*: Song et al., 2010), and therefore their presence in lieu of abundant conifers is not in
contradiction with an elevated topography in parts of east-central Tibet at this time. This has significance for
other Asian palynological studies that infer regional palaeoaltitudes and uplift history of Tibet based solely on
palynological records from a single locality: a multi-proxy approach is clearly necessary to address the complex
history of Tibetan uplift in future research.
Palynological data from the RZ assemblage supports climate (the third possibility) rather than altitude as a
primary driving factor of vegetational composition: locally wetter conditions in the east-central region of the TP
(see Section 5.2) would likely have promoted angiosperm tree growth over cold-temperate conifers that can
withstand drought better, and utilise a winter wet growing season unlike deciduous angiosperms (Dupont-Nivet
et al., 2008; Hoorn et al., 2012; Page et al., 2019). The last possibility is also supported, with the palynology of
this study suggesting that central Tibet was influenced by two atmospheric circulation systems: predominantly
the westerlies from the north (Caves Rugenstein and Chamberlain, 2018), and (to a limited degree) by a
southern monsoon, which could conceivably also have transported wind-blown pollen from sub-tropical and
warm temperate broad-leaved forests in the south (Su et al., 2018). Today, the Nangqian region receives nearly
70% of its moisture from the SAM, with the Westerlies from the north making up the remainder (Li L. et al.,
2019). This indicates that atmospheric circulation systems have changed considerably in east-central Tibet from
the Paleogene to Neogene, despite the existence of monsoons in this region since at least the Eocene (Licht et
al., 2014; Caves et al., 2017; Spicer, 2017). Based on the above, we propose that both local climatic conditions
and the influence of different regional atmospheric circulation systems contributed to the development of a
unique floral ecosystem in east-central Tibet during the late Eocene.

**6. Conclusions**
On the basis of palynological assemblages, we conclude that the rocks of the RZ section (Nangqian Basin)
are Bartonian (41.2–37.8 Ma; late Eocene) in age. They record a strongly seasonal steppe-desert ecosystem

characterised by *Ephedra* and *Nitraria* shrubs, diverse ferns and an underlying component of broad-leaved forests. The climate became significantly warmer for a short period, encouraging regional forest growth and a proliferation of the thermophilic ancestral *Ephedra* type, but rapidly aridified thereafter due primarily to regression of the proto-Paratethys Sea. This is in conjunction with observed environmental shifts in northeastern Tibet, suggesting widespread Asian aridification in the late Eocene. A new palynozonation better constrains the biostratigraphy of Paleogene successions across the northern, central, and southern TP, and also illustrates local ecological variability during the Eocene. This highlights the ongoing challenge of integrating various deep time records for the purpose of reconstructing palaeoelevation, and suggests that a multiproxy approach is vital for unravelling the complex uplift history of the Qinghai-Tibetan region.

**Author contribution**

Q.Y., V.V., F.S.S., D.L.G., H.C.W. and Q.S.F. conceptualized the study. Q.Y., F.S.S., H.C.W., Z.J.Q., Y.S.D. and J.J.S. carried out fieldwork. Q.Y., N.B., V.V. and C.R. collected and analysed the data. Q.Y. wrote the first draft and N.B., V.V., and C.R. participated in review and editing of the final draft.

**Competing interests**

The authors declare that they have no conflict of interest.

**Acknowledgments**

We thank Dr. Fuyuan An (Qinghai Normal University), Dr. Shuang Lü (University of Tübingen), and Aijun Sun (University of Chinese Academy of Sciences) for assistance with the fieldwork, Prof. Yunfa Miao (Chinese Academy of Sciences) for helpful discussions on the systematic palynology, Dr. Luisa Ashworth for lithological identifications, the MAGIC (Monsoons in Asia caused Greenhouse to Icehouse Cooling?) team for fruitful discussions on the topic, and the handling editor Prof. Alberto Reyes, as well as two anonymous reviewers, for their constructive comments. This work was supported by the National Natural Science Foundation of China (grant 41302024 to Q.Y.); The Youth Guiding Fund of Qinghai Institute of Salt Lakes, CAS (grant Y360391053 to Q.Y.); The Second Tibetan Plateau Scientific Expedition and Research Program (STEP) CAS (grant 2019 QZKK0805 to Q.Y.), and the Swedish Research Council (Vetenskapsrådet) grants VR 2019-4061 to V.V. and VR 2017-03985 to C.R. Funding sources had no involvement in study design.

**Data availability**
The authors declare that all data supporting the findings of this study are available in the supplementary
information or published in a data repository at the following DOI: http://dx.doi.org/10.17632/xvp68wsd2p.4.

**Supplementary information**
Supplementary information is available for this paper (Fig. S1, S2).

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

abstract).