# Peer review of "Aridification signatures from fossil pollen indicate a drying"

_Climate of the Past, 2019_

## Referee Comment (RC1) · Anonymous Referee #1 · 13 Feb 2020

The authors discussed the aridification on the Tibetan Plateau (TP) during the middle to late Eocene based on a palynological study from Nangqian Basin in northeastern TP. This work provides fundamental and important data for the evolution of plant diversity as well as paleoenvironmental change on the plateau. However, the authors fail to prove their conclusions with their data and explanations. Firstly, the authors need to clarify the position of Nangqian Basin on the TP, it seems that their statement is not consistent throughout the MS. In lines 50-51, it is "The uplifting, large-scale thrusting and striking of the TP caused several Paleogene intracontinental basins to form within the northern TP, including the Nangqian Basin"; but in lines 80-81, it is "The location of the Nangqian Basin on the east-central part of the TP". Moreover, there is few evidence

to indicate that the aridification in central Asia related to this northeastern part of the TP, actually, they belong to two different tectonic units. Therefore, it is beyond the scope of this study to use palynological evidence from northeastern TP to discuss the aridification of central Asia. Secondly, the authors need to use quantitative method (such as the pollen/spore percentage to evaluate if they might be in-situ or not) to discuss paleoelevation/paleoclimate in Nangqian Basin with palynologial data, because the downslope transport of pollen/spores from taxa living on high elevations could disturb their paleoenvironmental signals. Meanwhile, the authors should compare their results with recent studies from adjacent basins including Gonjo Basin and Markam Basin. Thirdly, I do not think that the geological age could be well constrained by palynological evidence such as Ephedra, which has quite rich fossil record throughout the Cenozoic. My another concern is the vegetation types concluded by the authors. How could the authors suggest a tropical forest in Zone II with data from few taxa? There should be much higher plant diversity and more thermophilic species in the assemblage if it is a real 'tropical forest'. Other minor suggestions: I suggest the authors to change the title. Why could the authors conclude a 'widespread' drying across the Tibetan Plateau mainly based on palynological study from one site in northeastern part of the plateau? The authors need to clarify it in the title even they have used data already published from different parts of the plateau in the analyses. Meanwhile, it is not accurate to use the word 'pollen', which only includes seed plants (angiosperm and gymnosperm). It is 'spore' in ferns, which the authors also observed in the sediment. The authors did not demonstrate on SEM method they used for taxonomic identification; moreover, they did not tell why only few pollen/spore s morphotypes were observed by SEM as shown in Plate III. Figure 1: The southeastern marginal part of Qiangtang Terrane should be much narrower than shown. Moreover, the authors need to uniform the format of cited references: few references are listed by full author names, and they are not in chronological order (e.g., Line 42); both 'and'/'&' (Line 321) occur in cited references.

---

## Referee Comment (RC2) · Anonymous Referee #2 · 2 Mar 2020

This manuscript, entitled, "Aridification signatures from middle-late Eocene pollen indicate widespread drying across the Tibetan Plateau after 40 Ma" by authors Yuan et al., presents a detailed and well-written new palynological study worth of publication in Climate of the Past. The new work on the RZ section from the Nangqian Basin may become a valuable contribution to the understanding of the climatic and tectonic histories of Tibet. This work does, however, require substantial revision in order to make a more compelling argument and to better communicate their findings to a broad audience. First, the authors should do a better job disclosing, both in the text as well as figures, where in the stratigraphic sections and to which zone each of the 21 productive samples belongs. For example, this should be clear for zone II, which the authors interpret

as MECO...are these interpretations based on a single sample? Such bold regional or global claims should be substantiated not only by robust evidence within the section but also corroborating evidence published elsewhere. I suggest the authors not only plot their samples on their stratigraphic sections (e.g. Figs 2, 3 and 4) but also discuss the statistical limitations of their samples (Zone II has only 2 samples; Zone III has 3). Further, I think the manuscript could benefit from additional discussion and a new figure similar to figure 3 that compares the palynological record presented here with non-palynological data such as stable isotope data from the region.

Second, there are ample opportunities to help this manuscript reach a broader audience. As a non-palynologist familiar with paleoclimate, I repeatedly found myself searching for the significance of some of the findings or the implications of a particular species abundance. This is particularly true for the paleoclimate discussion sections. For example: 1) Line 65: Explain the I-AM more; 2) Figure 1: These index maps aren't particularly useful. Perhaps something that is more (paleo)geographical or a vegetation map would help with the paleoclimate reconstructions to come?; 3) Figure 2: The ecological groups (e.g. Pteridophytes) could be better annotated for non-specialists, NLR should be explained, and N/E ratios could be labeled desert/semi-desert and steppe-desert; 4) Figure 3: Index map could be greatly improved and this study could be highlighted with a different marker. The plant functional types listed here aren't being consistently used throughout the paper (e.g. "temperate broad-leaved forest" etc in figure 2). These should be consistent throughout; 5) Figure 4: These taxa should be explained, especially as you go on to stress the importance of Ef/Ed ratios later; 6) Background on MECO should be developed earlier; 7) PFTs should be developed earlier and consistent throughout the text; 8) More explanation is needed as to why you favor N/E over Ef/Ed; 9) Age constraints should include Ma throughout in addition to just stratigraphic stages e.g. line 423.

---

## Author Comment (AC1) · 2 Apr 2020

1) The authors discussed the aridification on the Tibetan Plateau (TP) during the middle to late Eocene based on a palynological study from Nangqian Basin in northeastern TP. This work provides fundamental and important data for the evolution of plant diversity as well as paleoenvironmental change on the plateau.

We would like to thank the reviewer for their time in reviewing our manuscript, and for their insightful comments which have helped to improve the work.

2) Firstly, the authors need to clarify the position of Nangqian Basin on the TP, it seems

that their statement is not consistent throughout the MS. In lines 50-51, it is "The uplifting, large-scale thrusting and striking of the TP caused several Paleogene intracontinental basins to form within the northern TP, including the Nangqian Basin"; but in lines 80-81, it is "The location of the Nangqian Basin on the east-central part of the TP".

We agree and will ensure this is made consistent in the revised manuscript.

3) There is few evidence to indicate that the aridification in central Asia related to this northeastern part of the TP, actually, they belong to two different tectonic units. Therefore, it is beyond the scope of this study to use palynological evidence from northeastern TP to discuss the aridification of central Asia.

While uplift of the TP has traditionally been invoked to explain the onset of Asian aridification, retreat of the proto-Paratethys Sea in the Eocene has now also been shown as a major factor (Kaya et al., 2019). This sea extended from the Mediterranean Tethys to the Tarim Basin in western China, and through moisture transport via the westerlies, constituted a major moisture source to the Central Asian interior (Bougeois, 2014; Bougeois et al., 2018; Caves et al., 2015) despite its eastern extent being thousands of kilometres (roughly equidistant) from both the Xining and Nangqian basins. Both Northern Tibet (Xining Basin) and Central Asia (east-central Tibet: Nangqian Basin) have received moisture dominantly via the westerlies, which have maintained a semi-arid to arid climate in Central Asia since the early Eocene (Caves Rugenstein & Chamberlain, 2018). Therefore, we argue that aridification in both parts of Tibet is indeed related to a single, large-scale atmospheric transport system operating over this part of the TP during the Eocene, which justifies our comparison of palynological records.

An additional reason we use NE Tibet for more detailed comparison and age correlation is that this is one of the few sections on the TP that is both time-extended (Paleocene-Oligocene; Dupont-Nivet et al., 2008, 2008; Hoorn et al., 2012; Bosboom et al., 2014) and has good independent age control. This allows to observe long-term trends in palynomorph variation through time, so that correlations between different sections

can be based on real vegetation changes instead of possible short-term fluctuations that would not be detected in less time-extensive sections. We do caution in Lines 334-335 that further investigations should be made in Nangqian and other parts of the Tibetan Plateau using independent age control to corroborate this finding. However, we support our results with biostratigraphic correlation to multiple other assemblages from different parts of the TP (illustrated in Figs. 1 & 3) in order to detect large-scale patterns of vegetation change across the plateau through the Paleogene. While there are obviously regional controls on plant distributions and abundance on different parts of the plateau, there are also broad similarities that can be used for correlation between all of the sections, which was previously recognised by Wang et al., 1990a, 1990b, Sun & Wang, 2005, and others. We now also include the adjacent basins of Gonjo and Markam for comparison as suggested in a later comment.

4) The authors need to use quantitative method (such as the pollen/spore percentage to evaluate if they might be in-situ or not) to discuss paleoelevation/ paleoclimate in Nangqian Basin with palynologial data, because the downslope transport of pollen/spores from taxa living on high elevations could disturb their paleoenvironmental signals.

Palynological assemblages generally reflect the regional vegetation, except in particular environmental settings such as coal swamps in which the autochthonous palynomorph content can be up to 100%. Therefore, it is expected that the assemblage will not only record vegetation that was present at the site itself but also the wider area, and this is beneficial as it reflects regional climate instead of conditions that could be locally controlled. By the middle-late Eocene on the TP, it becomes clear that palynological assemblages reflect a vertical zonation of vegetation, and the existence of surrounding higher elevations (e.g., Hoorn et al., 2012; Wu et al., 2018). Accordingly, many significant works on the palaeoelevation and palaeoclimates on the TP (e.g., Song et al. 2010; Hoorn et al., 2012; Miao et al., 2013; Sun et al. 2007, 2008; Miao at el., 2016; Wu et al. 2018) have used palynology without applying quantitative methods to determine whether all the palynomorphs were deposited at site, because it is already known this is not the case. In recognition of the fact that the assemblage reflects the regional vegetation, we do not use palynology to calculate precise climatic parameters such as mean annual temperature or mean annual precipitation. In Section 5.3: Elevational implications, we discuss in detail the difficulties of calculating palaeoelevation based solely on palynology, including the issue of arboreally transported pollen. Because of this factor, we did not use conifers or other taxa prone to longer-distance transport (e.g., Alnus, Betula) to estimate palaeoclimatic conditions, in order to be cautious about allowing taxa from a potentially significantly different elevation to influence the paleoenvironmental analysis. In our section the percentage of spores is high relative to other TP basins (Miao at el., 2016), suggesting a significant proportion of deposition at site, and we will insert discussion on this into the revised manuscript.

5) The authors should compare their results with recent studies from adjacent basins including Gonjo Basin and Markam Basin.

We agree and will include comparisons to these basins in the revised manuscript; regarding pollen data from the Gonjo Basin, at present we are only aware of two publications: 1) "BGMRX, 1993. Regional Geology of Xizang (Tibet) Autonomous Region. Geol. Mem., vol.1. Geological Publishing House, Beijing" which contains a short mention of some species from the tops of the Gongjue Formation and Lawula Group but unfortunately no percentage data or information about the Ranmugou Formation; and 2) "Studnicki-Gizbert, C., Burchfiel, B.C., Li, Z. and Chen, Z., 2008. Early Tertiary Gonjo basin, eastern Tibet: Sedimentary and structural record of the early history of India-Asia collision. Geosphere, 4(4), 713-735" which reports only 3 very poorly preserved palynological samples, making it difficult to correlate with our section. If the reviewer could let us know of any further references that exist with more information on pollen content from Gonjo Basin, this would be appreciated.

6) I do not think that the geological age could be well constrained by palynological evidence such as Ephedra, which has quite rich fossil record throughout the Cenozoic.

The age of our section has been determined using a variety of different constraints, including K-Ar ages, zircon U-Pb age data, and biostratigraphic means. Firstly, emplacement ages from shoshonitic lavas and felsic and porphyry intrusions that are either interbedded with, or unconformably overlie, the lacustrine to alluvial Nangqian strata were used to determine a minimum age of ca. 37-38 Ma for the RZ section. This is congruent with palynological evidence for the overall age of the sampled strata. Next, biostratigraphic correlation between assemblage from the RZ section and other parts of the Tibetan Plateau (Fig. 3) provides a refinement of the age to middle-late Eocene. This is discussed in detail in Section 5.1: Age assignment in the main manuscript. The relative abundances of Ephedripites (Ephedripites) and Ephedripites (Distachyapites) are further proposed to constrain the age, as at some point during the Paleogene Ephedripites (Distachyapites) became more abundant than Ephedripites (Ephedripites), which is common in the Cretaceous (Han et al., 2016; Bolinder et al., 2016). In NE Tibet this change has been determined to be from ca. 39 Ma onwards, but we agree that this may not have occurred across the TP simultaneously. Expanded discussion on this will be included in the revised manuscript to justify use of this approach. Furthermore, we agree that it is challenging to determine a precise age from palynology, and hence we will adopt a more cautious approach by revising our final assignment to an age range (late Lutetian-Bartonian) for our section rather than a specific age (i.e., 42 Ma; 40 Ma/MECO; 38 Ma as in Fig. 5) for each of the pollen zones.

7) How could the authors suggest a tropical forest in Zone II with data from few taxa? There should be much higher plant diversity and more thermophilic species in the assemblage if it is a real 'tropical forest'.

We agree, and recognize that this implication should be better discussed in the revised manuscript. As pointed out by the reviewer, the palynology does not indicate the existence of a real 'tropical forest' during Zone II but only an increase in regional input of some tropical taxa. While this suggests a temporary warming period, it does not mean a complete biome transition from steppe vegetation to forest. In the original manuscript

this was not made clear, and we will amend this. We are however confident that this zone is distinct and represents a change in climate based on three lines of evidence. Firstly, this zone shows a large decrease in steppe-desert pollen which is not observed in the other zones of this section (average 9% steppe-desert pollen in Zone II vs 38% (Zone I) and 32 % (Zone III)), nor later in the Eocene in the Nangqian Basin (Yuan et al., 2017). There is also a spike in the ancestral Ephedra type during Zone II, and this is also not observed elsewhere in this section or that of Yuan et al. (2017). This spike in ancestral Ephedra, together with an increase in warm forest, are only observed over the MECO in the Xining Basin, NE Tibet and not later in the Eocene (Hoorn et al., 2012; Han et al., 2016) or in the middle Eocene (Meijer et al., submitted). Lastly, the tropical forest spike in Zone II of the RZ section is unusual and also not observed elsewhere in this section or elsewhere in Nangqian in the Eocene (Yuan et al., 2017) or the late Paleocene-early Eocene of Nangqian (Barbolini et al. 2018: Barbolini, unpublished data), however we recognise that this spike is only present in one sample, and therefore further investigations should be made in Nangqian and other parts of the Tibetan Plateau to corroborate this finding. We mention this limitation in the Discussion section (lines 318–325). We are also confident that the pollen in Zone II do not represent reworking or contamination, as the palynomorphs from these samples were not degraded or compressed to a greater degree than palynomorphs from the rest of the section, and of a similar colour and appearance. However, we will also include a discussion on statistical limitations of the samples.

8) I suggest the authors to change the title. Why could the authors conclude a 'widespread' drying across the Tibetan Plateau mainly based on palynological study from one site in northeastern part of the plateau? The authors need to clarify it in the title even they have used data already published from different parts of the plateau in the analyses. It is not accurate to use the word 'pollen', which only includes seed plants (angiosperm and gymnosperm). It is 'spore' in ferns, which the authors also observed in the sediment.

We are aware that both spores and pollen are present in the samples as illustrated in Plates I-III, however the use of the word "pollen" in the title refers to the progressive aridification observed from key pollen species in the samples, and therefore its use is accurate in that case. Throughout the main manuscript, we will ensure that when the term "pollen" could also include spores, this shall be changed to "palynological" to avoid ambiguity. We agree the title should be modified and contracted to focus on the present results, i.e. "Aridification signatures from middle-late Eocene pollen indicate a drying climate on the east-central Tibetan Plateau".

9) The authors did not demonstrate on SEM method they used for taxonomic identification; moreover, they did not tell why only few pollen/spore s morphotypes were observed by SEM as shown in Plate III.

As is standard for palynostratigraphic studies, we used primarily LM (light microscopy) to identify, count, and photograph the pollen and spores present in the samples (e.g., Traverse, A., 2007, Paleopalynology, 2nd ed. Springer, Dordrecht, Appendix: Palynological Laboratory Techniques and p. 53: "light microscopy is the workhorse method for study of palynomorphs, and this will remain the case for the immediate future"). The SEM plate is included primarily to illustrate the appearances of Ephedripites (Ephedripites) and Ephedripites (Distachyapites) under SEM as well as some other key species in different palynozones of the studied section. SEM was not necessary for taxonomic identifications of all of the pollen and spores present, thus duplicate SEM plates showing the same palynomorphs as Plates I and II were not included. This explanation will be added to the Methods section for clarity.

10) Figure 1: The southeastern marginal part of Qiangtang Terrane should be much narrower than shown.

We agree; this will be amended and the Songpan-Ganzi Terrane also marked above.

11) The authors need to uniform the format of cited references: a few references are listed by full author names, and they are not in chronological order (e.g., Line 42); both

'and'/'&' (Line 321) occur in cited references.

The references have been double-checked for consistency and errors amended.

---

## Author Comment (AC2) · 2 Apr 2020

A) This manuscript, entitled, "Aridification signatures from middle-late Eocene pollen indicate widespread drying across the Tibetan Plateau after 40 Ma" by authors Yuan et al., presents a detailed and well-written new palynological study worth of publication in Climate of the Past. The new work on the RZ section from the Nangqian Basin may become a valuable contribution to the understanding of the climatic and tectonic histories of Tibet.

We would like to thank the reviewer for their positive evaluation of our manuscript, and for their insightful comments which have helped to improve the work.

[Figure]

B) First, the authors should do a better job disclosing, both in the text as well as figures, where in the stratigraphic sections and to which zone each of the 21 productive samples belongs. For example, this should be clear for zone II, which the authors interpret as MECO: are these interpretations based on a single sample? Such bold regional or global claims should be substantiated not only by robust evidence within the section but also corroborating evidence published elsewhere. I suggest the authors not only plot their samples on their stratigraphic sections (e.g. Figs 2, 3 and 4) but also discuss the statistical limitations of their samples (Zone II has only 2 samples; Zone III has 3).

We agree and will expand on this in the revised manuscript: based on comments from both reviewers we have decided to adopt a more cautious approach to our age assignment due to the limited number of samples, and will assign an age range (late Lutetian-Bartonian) to our section rather than a specific age (i.e., 42 Ma; 40 Ma/MECO; 38 Ma as in Fig. 5) to each of the pollen zones. We are confident that the palynological character of the assemblages combined with the K-Ar ages and zircon U-Pb age data (discussed in Section 5.1: Age assignment) is sufficient evidence for assigning this age range. There are three different lines of evidence that support this age assignment (lines 296-325). Firstly, this zone shows a large decrease in steppe-desert pollen which is not observed in the other zones of this section (average 9% steppe-desert pollen in Zone II vs 38% (Zone I) and 32 % (Zone III)), nor later in the Eocene in the Nangqian Basin (Yuan et al., 2017). There is also a spike in the ancestral Ephedra type during Zone II, and this is also not observed elsewhere in this section or that of Yuan et al. (2017). This spike in ancestral Ephedra, together with an increase in warm forest, are only observed over the MECO in the Xining Basin, NE Tibet and not later in the Eocene (Hoorn et al., 2012; Han et al., 2016) or in the middle Eocene (Meijer et al., submitted). Lastly, the tropical forest spike in Zone II of the RZ section is unusual and also not observed elsewhere in this section or elsewhere in Nangqian in the Eocene (Yuan et al., 2017) or the late Paleocene-early Eocene of Nangqian (Barbolini et al. 2018: Barbolini, unpublished data), however we recognise that this spike is only present in one sample, and therefore further investigations should be made in Nangqian and other

parts of the Tibetan Plateau to corroborate this finding. We mention this limitation in the Discussion section (lines 318-325). We are also confident that the pollen in Zone II do not represent reworking or contamination, as the palynomorphs from these samples were not degraded or compressed to a greater degree than palynomorphs from the rest of the section, and of a similar colour and appearance. However, we will also include a discussion here on statistical limitations of the samples. These are already plotted on Fig. 2 and we will also plot them on Fig. 4; to do this on Fig. 3 is challenging because of the reduced interval of time the studied section encompasses compared to other sections across the TP. Expanding the figure to allow 21 samples to be plotted on Section 4 (this study) would render the figure too large for publication.

C) Further, I think the manuscript could benefit from additional discussion and a new figure similar to figure 3 that compares the palynological record presented here with non-palynological data such as stable isotope data from the region.

Unfortunately, we did not obtain stable isotope data during our study and generating a new figure on this spanning the TP is outside the scope of this study, but our record can be compared in the text with previous studies presenting these data from the Nangqian Basin, e.g., Li et al., 2019. Carbonate stable and clumped isotopic evidence for late Eocene moderate to high elevation of the east-central Tibetan Plateau and its geodynamic implications. GSA Bulletin, 131(5-6), 831-844.

D) Second, there are ample opportunities to help this manuscript reach a broader audience. As a non-palynologist familiar with paleoclimate, I repeatedly found myself searching for the significance of some of the findings or the implications of a particular species abundance. This is particularly true for the paleoclimate discussion sections. For example: 1) Line 65: Explain the I-AM more.

We agree and will expand on this in the revised manuscript.

2) Figure 1: These index maps aren't particularly useful. Perhaps something that is more (paleo)geographical or a vegetation map would help with the paleoclimate

reconstructions to come?

We agree and will replace Fig. 1A with an Eocene palaeogeography of the area with locations of the different basins marked. Fig. 1B will be a current vegetation distribution map of the TP to allow for comparison with the reconstructed Eocene vegetation presented later.

3) Figure 2: The ecological groups (e.g. Pteridophytes) could be better annotated for non-specialists, NLR should be explained, and N/E ratios could be labeled desert/semi-desert and steppe-desert.

We agree and will clarify these points in the revised manuscript.

4) Figure 3: Index map could be greatly improved and this study could be highlighted with a different marker. The plant functional types listed here aren't being consistently used throughout the paper (e.g. "temperate broad-leaved forest" etc in figure 2). These should be consistent throughout.

As above, we will replace the index map, as well as highlight this study and standardise terminology of the plant functional types.

5) Figure 4: These taxa should be explained, especially as you go on to stress the importance of Ef/Ed ratios later.

We agree and will expand on this in the revised manuscript.

6) Background on MECO should be developed earlier.

We agree and will amend this in the revised manuscript.

7) PFTs should be developed earlier and consistent throughout the text.

We agree and will amend this in the revised manuscript.

8) More explanation is needed as to why you favor N/E over Ef/Ed.

We agree and will expand on this in the revised manuscript.

9) Age constraints should include Ma throughout in addition to just stratigraphic stages e.g. line 423.

We agree and will amend this in the revised manuscript.

---

## Author Response (AR1)

June 2020

Editorial Board
Climate of the Past

Dear Prof. Alberto Reyes

Reply to reviewer comments on MS No.: cp-2019-138

We thank both reviewers for their time and effort in reviewing our manuscript "Aridification signatures from fossil pollen indicate a drying climate in east-central Tibet during the late Eocene". Below we provide responses to their comments with line numbers indicating where we have made changes, and include a separate marked-up manuscript version below showing the changes made.

*Please be advised, line numbers below refer to lines in the marked up version of the document, not the clean copy.*

Review: Anonymous Referee #1

*1. The authors discussed the aridification on the Tibetan Plateau (TP) during the middle to late Eocene based on a palynological study from Nangqian Basin in northeastern TP. This work provides fundamental and important data for the evolution of plant diversity as well as paleoenvironmental change on the plateau.*

We would like to thank the reviewer for their time in reviewing our manuscript, and for their insightful comments which have helped to improve the work.

*2. Firstly, the authors need to clarify the position of Nangqian Basin on the TP, it seems that their statement is not consistent throughout the MS. In lines 50-51, it is "The uplifting, large-scale thrusting and striking of the TP caused several Paleogene intracontinental basins to form within the northern TP, including the Nangqian Basin"; but in lines 80-81, it is "The location of the Nangqian Basin on the east-central part of the TP".*

We apologise for the error and have ensured this is consistent in the revised manuscript: in lines 84-85 we have revised the sentence to "The uplifting, large-scale thrusting and striking of Tibet caused several Paleogene intracontinental basins to form within the northern and central Qinghai-Tibetan region, including the Nangqian Basin".

*3. There is few evidence to indicate that the aridification in central Asia related to this northeastern part of the TP, actually, they belong to two different tectonic units. Therefore, it is beyond the scope of this study to use palynological evidence from northeastern TP to discuss the aridification of central Asia.*

We appreciate the comment of the reviewer. While uplift of the TP has traditionally been invoked to explain the onset of Asian aridification, retreat of the proto-Paratethys Sea in the Eocene has now also been shown as a major factor (Kaya et al., 2019). This sea extended from the Mediterranean Tethys to the Tarim Basin in western China, and through moisture transport via the westerlies, constituted a major moisture source to the Central Asian interior (Bougeois, 2014; Bougeois et al., 2018; Caves et al., 2015) despite its eastern extent being thousands of kilometres (roughly equidistant) from both the Xining and Nangqian basins.

Both Northern Tibet (Xining Basin) and Central Asia (east-central Tibet: Nangqian Basin) have received moisture dominantly via the westerlies, which have maintained a semi-arid to arid climate in Central Asia since the early Eocene (Caves Rugenstein & Chamberlain, 2018). Therefore, we argue that aridification in both parts of Tibet is indeed related to a single, large-scale atmospheric transport system operating over this part of the TP during the Eocene, which justifies our comparison of palynological records. We have now inserted further discussion on this in lines 98-103.

An additional reason we use NE Tibet for more detailed comparison and age correlation is that this is one of the few sections on the TP that is both time-extended (Paleocene–Oligocene; Dupont-Nivet et al., 2008, 2008; Hoorn et al., 2012; Bosboom et al., 2014) and has good independent age control throughout. We now discuss this in lines 359-361. This allows to observe long-term trends in palynomorph variation through time, so that correlations between different sections can be based on real vegetation changes instead of possible short-term fluctuations that would not be detected in less time-extensive sections.

*4. The authors need to use quantitative method (such as the pollen/spore percentage to evaluate if they might be in-situ or not) to discuss paleoelevation/ paleoclimate in Nangqian Basin with palynologial data, because the downslope transport of pollen/spores from taxa living on high elevations could disturb their paleoenvironmental signals.*

Palynological assemblages generally reflect the regional vegetation, except in particular environmental settings such as coal swamps in which the autochthonous palynomorph content can be up to 100% (Traverse, 2007). Therefore, it is expected that the assemblage will not only record vegetation that was present at the site itself but also the wider area, and this is beneficial as it reflects regional climate instead of conditions that could be locally controlled. By the middle-late Eocene on the TP, it becomes clear that palynological assemblages reflect a vertical zonation of vegetation, and the existence of surrounding higher elevations (e.g., Hoorn et al., 2012; Wu et al., 2018).

Accordingly, many significant works on the palaeoelevation and palaeoclimates on the TP (e.g., Song et al. 2010; Hoorn et al., 2012; Miao et al., 2013; Sun et al. 2007, 2008; Miao at el., 2016; Wu et al. 2018) have used palynology without applying quantitative methods to determine whether all the palynomorphs were deposited at site, because it is already known this is not the case. In our section the percentage of spores is high relative to other TP basins (see Miao at el., 2016 for a synthesis of previous records), suggesting a significant proportion of deposition at site, and we have inserted discussion on this into the revised manuscript in lines 234-241. We also reference studies indicating that pollen assemblages provide a good reconstruction of the regional vegetation, which is our aim.

In recognition of the fact that our assemblage reflects the regional vegetation, we do not use palynology to calculate precise climatic parameters such as mean annual temperature or mean annual precipitation. We are also cautious about allowing taxa from a potentially significantly different elevation to influence the paleoenvironmental analysis. In Section 5.3: Elevational implications, we discuss in detail the difficulties of calculating palaeoelevation based solely on palynology, including the issue of arboreally transported pollen. Because of this factor, we did not use conifers or other taxa prone to longer-distance transport (e.g., *Alnus, Betula*) to estimate palaeoclimatic conditions.

*5. The authors should compare their results with recent studies from adjacent basins including Gonjo Basin and Markam Basin.*

We agree with the point of the reviewer. We included a discussion of recent stable isotope data from the Gonjo Basin (Tang et al., 2017; lines 555-558) and macrobotanical remains from the Markam Basin (Su et al., 2018; lines 527-530).

Regarding palynological data from both of these basins, unfortunately there are not yet detailed enough records that allow for comparison with our palynological section. Macrobotanical remains are available from the Eocene Markam Basin (Su et al., 2018), but palynological records are Miocene: J.-R. Tao, N.-Q. Du, Miocene flora from Markam County and fossil record of Betulaceae. Acta Bot. Sin. 29, 649– 655 (1987).

We are aware of two publications from the Gonjo Basin referencing pollen:
1) "BGMRX, 1993. Regional Geology of Xizang (Tibet) Autonomous Region. Geol. Mem., vol.1. Geological Publishing House, Beijing" which contains a short mention of some species from the tops of the Gongjue Formation and Lawula Group but unfortunately no percentage data or information about the Ranmugou Formation; and
2) "Studnicki-Gizbert, C., Burchfiel, B.C., Li, Z. and Chen, Z., 2008. Early Tertiary Gonjo basin, eastern Tibet: Sedimentary and structural record of the early history of India-Asia collision. Geosphere, 4(4), 713-735" which reports only 3 very poorly preserved palynological samples.

We now reference these studies in the text, and did our best to add other relevant records from adjacent basins. We also present a new section from the Jianchuan Basin in Fig. 3, which provides an additional comparison from the southern part of the plateau.

*6. I do not think that the geological age could be well constrained by palynological evidence such as Ephedra, which has quite rich fossil record throughout the Cenozoic.*

Indeed, the record of *Ephedripites* pollen is particularly good in the Cenozoic of Asia. However, the age of our section has been determined using a variety of different constraints, including K–Ar ages, zircon U–Pb age data, and biostratigraphic means. This is discussed in detail in Section 5.1: Age assignment in the main manuscript.

Firstly, emplacement ages from shoshonitic lavas and felsic and porphyry intrusions that are either interbedded with, or unconformably overlie, the lacustrine to alluvial Nangqian strata were used to determine a minimum age of ~37–38 Ma for the RZ section. This is congruent with palynological evidence for the overall age of the sampled strata (lines 329-341).

Next, biostratigraphic correlation between assemblage from the RZ section and other parts of the Tibetan Plateau (Fig. 3) provides a refinement of the age to middle–late Eocene (lines 342-353).

Subsequently, we propose that the relative abundances of *Ephedripites (Ephedripites)* and *Ephedripites (Distachyapites)* can further constrain the age to late Eocene (Bartonian), as at this point *Ephedripites (Distachyapites)* became more abundant than *Ephedripites (Ephedripites)*, which is common in the Cretaceous (Han et al., 2016; Bolinder et al., 2016). In NE Tibet this change has been determined to be from ~39 Ma onwards, but we agree that this may not have occurred across the TP simultaneously. Expanded discussion on this is now included in the revised manuscript to justify use of this approach (lines 367-370).

Furthermore, we agree that it is challenging to determine a precise age from palynology, and hence we adopt a more cautious approach by assigning an age range (Bartonian) for our section rather than a specific age for each of the pollen zones.

*7. How could the authors suggest a tropical forest in Zone II with data from few taxa? There should be much higher plant diversity and more thermophilic species in the assemblage if it is a real 'tropical forest'.*

The reviewer makes a valuable point. We apologise for not making this clear in our original text. As pointed out by the reviewer, the palynology does not indicate the existence of a real 'tropical forest' during Zone II but only an increase in regional input of some tropical taxa. While this suggests a temporary warming period, it does not mean a complete biome transition from steppe vegetation to forest. We have modified the text to reflect this in lines 316 and 387-389.

Current evidence however suggests that this zone is distinct and represents a change in regional climate. We have now modified the text in lines 406-415 to make this clearer:

Firstly, this zone shows a large decrease in steppe-desert pollen which is not observed in the other zones of this section (average 9% steppe-desert pollen in Zone II vs 38% (Zone I) and 32 % (Zone III)), nor later in the Eocene in the Nangqian Basin (Yuan et al., 2017).

There is also a spike in the ancestral *Ephedra* type during Zone II, and this is also not observed elsewhere in this section or that of Yuan et al. (2017). This spike in ancestral *Ephedra*, together with an increase in warm forest, are only observed between 41-39 Ma in the Xining Basin, NE Tibet (Hoorn et al., 2012; Han et al., 2016).

We are also confident that the pollen in Zone II do not represent reworking or contamination, as the palynomorphs from these samples were not degraded or compressed to a greater degree than palynomorphs from the rest of the section, and of a similar colour and appearance.

Lastly, the tropical forest spike in Zone II of the RZ section is unusual and also not observed elsewhere in this section or elsewhere in Nangqian in the Eocene (Yuan et al., 2017) or the late Paleocene–early Eocene of Nangqian (Barbolini et al. 2018: Barbolini, unpublished data), however we recognise that this spike is only present in one sample, and therefore further investigations should be made in Nangqian and other parts of the Tibetan Plateau to corroborate this finding. We discuss this limitation in the Discussion section (lines 398-406).

*8. I suggest the authors to change the title. Why could the authors conclude a 'widespread' drying across the Tibetan Plateau mainly based on palynological study from one site in northeastern part of the plateau? The authors need to clarify it in the title even they have used data already published from different parts of the plateau in the analyses. It is not accurate to use the word 'pollen', which only includes seed plants (angiosperm and gymnosperm). It is 'spore' in ferns, which the authors also observed in the sediment.*

We are aware that both spores and pollen are present in the samples as illustrated in Plates I–III, however the use of the word "pollen" in the title refers to the progressive aridification observed from key pollen species in the samples, and therefore its use is accurate in that case. Throughout the main manuscript, we have ensured that when the term "pollen" could also include spores, this has been changed to "palynological" to avoid ambiguity.

We agree the title should be modified and contracted to focus on the present results. It has now been changed to "Aridification signatures from fossil pollen indicate a drying climate in east-central Tibet during the late Eocene".

*9. The authors did not demonstrate on SEM method they used for taxonomic identification; moreover, they did not tell why only few pollen/spore s morphotypes were observed by SEM as shown in Plate III.*

As is standard for palynostratigraphic studies, we used primarily LM (light microscopy) to identify, count, and photograph the pollen and spores present in the samples (e.g., Traverse, A., 2007, Paleopalynology, 2nd ed. Springer, Dordrecht, Appendix: Palynological Laboratory Techniques and p. 53: "light microscopy is the workhorse method for study of palynomorphs, and this will remain the case for the immediate future").

The SEM plate is included primarily to illustrate the appearances of *Ephedripites (Ephedripites)* and *Ephedripites (Distachyapites)* under SEM as well as some other key species in different palynozones of the studied section. SEM was not necessary for taxonomic identifications of all of the pollen and spores present, thus duplicate SEM plates showing the same palynomorphs as Plates I and II were not included. This explanation has been added to the Methods section for clarity (lines 201-204).

*10. Figure 1: The southeastern marginal part of Qiangtang Terrane should be much narrower than shown.*

We apologise for this error; this has been amended and the Songpan-Ganzi Terrane also marked in a redrawn geological map for Fig. 1.

*11. The authors need to uniform the format of cited references: a few references are listed by full author names, and they are not in chronological order (e.g., Line 42); both 'and'/'&' (Line 321) occur in cited references.*

The references have been double-checked for consistency and these errors amended.

Review: Anonymous Referee #2

*1. This manuscript, entitled, "Aridification signatures from middle-late Eocene pollen indicate widespread drying across the Tibetan Plateau after 40 Ma" by authors Yuan et al., presents a detailed and well-written new palynological study worth of publication in Climate of the Past. The new work on the RZ section from the Nangqian Basin may become a valuable contribution to the understanding of the climatic and tectonic histories of Tibet.*

We would like to thank the reviewer for their positive evaluation of our manuscript, and for their insightful comments which have helped to improve the work.

*2. First, the authors should do a better job disclosing, both in the text as well as figures, where in the stratigraphic sections and to which zone each of the 21 productive samples belongs. For example, this should be clear for zone II, which the authors interpret as MECO: are these interpretations based on a single sample? Such bold regional or global claims should be substantiated not only by robust evidence within the section but also corroborating evidence published elsewhere. I suggest the authors not only plot their samples on their stratigraphic sections (e.g. Figs 2, 3 and 4) but also discuss the statistical limitations of their samples (Zone II has only 2 samples; Zone III has 3).*

We agree with this point; based on comments from both reviewers we have decided to adopt a more cautious approach to our age assignment due to the limited number of samples, and have assigned an age range (Bartonian; 41.2–37.8 Ma) to our section rather than a specific age to each of the pollen zones. We now mention explicitly that Zone II contains only 2 samples and Zone III has 3 in lines 398-407, and this places statistical limitations on our interpretations. We further mention that a correlation to the MECO cannot be confidently made on the basis of this single sample, and thus we do not date the individual pollen zones (lines 406-407). Rather, the palynological evidence, together with the K–Ar ages and zircon U–Pb age data, indicates a Bartonian (41.2–37.8 Ma) age for the section (lines 421-424).

However, we suggest that Zone II represents a regional change in climate based on the spike in thermophilic pollen, the large decrease in steppe-desert pollen, and a spike in the ancestral *Ephedra* type, all of which can also be observed in northeast Tibet in the Bartonian, thus it appears there is a change at this time (lines 406-415).

In the original manuscript the productive samples were plotted on Fig. 2, and we now also do this on Fig. 4. To do this on Fig. 3 is challenging because of the reduced interval of time the studied section encompasses compared to other sections across the TP. Expanding the figure to allow 21 samples to be plotted on the studied section would render the figure too large for publication.

*3. Further, I think the manuscript could benefit from additional discussion and a new figure similar to figure 3 that compares the palynological record presented here with non-palynological data such as stable isotope data from the region.*

Unfortunately, we did not obtain stable isotope data during our study and generating a new figure on this spanning the TP is outside the scope of this study, but our record is now compared in the text with recent studies presenting these data from the Nangqian Basin (Li et al., 2019, GSA Bulletin; lines 483, 555-558, 574-575) and the Gonjo Basin (Tang et al., 2017; lines 557-558).

*4. Second, there are ample opportunities to help this manuscript reach a broader audience. As a non-palynologist familiar with paleoclimate, I repeatedly found myself searching for the significance of some of the findings or the implications of a particular species abundance. This is particularly true for the paleoclimate discussion sections. For example:*

*1) Line 65: Explain the I-AM more.*

We insert discussion on this now in lines 108-111 and 464-466, highlighting how monsoonal circulation in central Tibet has changed since the Paleogene.

*2) Figure 1: These index maps aren't particularly useful. Perhaps something that is more (paleo)geographical or a vegetation map would help with the paleoclimate reconstructions to come?*

The reviewer makes a valuable point. We have redrawn Fig. 1 to incorporate A. a geological map indicating structural features and present altitude, B. an Eocene palaeogeography of the area, and C. a current vegetation distribution map of the Tibetan Plateau which allows for comparison with the reconstructed Eocene vegetation presented later (now discussed in lines 521-533)

*3) Figure 2: The ecological groups (e.g. Pteridophytes) could be better annotated for non-specialists, NLR should be explained, and N/E ratios could be labeled desert/semi-desert and steppe-desert.*

We agree this could have been better explained. We have now annotated Fig. 2 with Pteridophytes (ferns) in the legend and explained the Plant Functional Types (PFTs)and Nearest Living Relatives (NLR) in the figure caption. We tried labelling the N/E ratios as desert/semi-desert and steppe-desert directly in the figure, but found that it looked very squashed and seemed confusing. We rather explain this in the figure caption.

*4) Figure 3: Index map could be greatly improved and this study could be highlighted with a different marker. The plant functional types listed here aren't being consistently used throughout the paper (e.g. "temperate broad-leaved forest" etc in figure 2). These should be consistent throughout.*

We have replaced the index map with a map illustrating the present vegetation distribution across the Tibetan Plateau. We now also mark on each basin the dominant vegetation that existed through the Cenozoic, and discuss how the vegetation has changed over time in the Qinghai-Tibetan region (lines 521-533).

We highlighted our study with a yellow box to match the map location. We also standardised terminology of the plant functional types (changed in lines 302-304, 307, 308-309, 317, 318, 324, 432, 438, 470, 480) to ensure consistency with Figs. 2, 3 and 5.

*5) Figure 4: These taxa should be explained, especially as you go on to stress the importance of Ef/Ed ratios later.*

We annotated Fig 4 to better show the ancestral (Ef) and derived (Ed) types, and now explain this in the figure caption as well (lines 381-384).

*6) Background on MECO should be developed earlier.*

We now introduce the MECO in the Introduction (lines 61-67) first, and then refer to it again in the Discussion (lines 389-394).

*7) PFTs should be developed earlier and consistent throughout the text.*

We now describe our approach using Plant Functional Types (PFTs) in the Methods section first (lines 207-209) and then again in the caption of Fig. 2 (lines 247-249). We have amended wording in lines 302-304, 307, 308-309, 317, 318, 324, 432, 438, 470, and 480 to ensure consistency with Figs. 2, 3 and 5.

*8) More explanation is needed as to why you favor N/E over Ef/Ed.*

We now explain this in lines 514-516.

*9) Age constraints should include Ma throughout in addition to just stratigraphic stages e.g. line 423.*

We have amended this in lines 22, 345, 370, 423 and 586.

[revised manuscript text omitted]
 Nangqian Basin. (B) late–middle Eocene (40 Ma) palaeogeographic reconstruction with the Qinghai-Tibetan region indicated by a black rectangle (redrawn after Tardif et al., 2020). (C) Modern vegetation distributions on the Tibetan Plateau with major towns indicated in red (redrawn after Baumann et al., 2009). Numbers indicate the positions of palynological assemblages that are correlated in Fig. 3 and the text: 1. Tarim Basin; 2. Hoh Xil Basin; 3, 4. Nangqian Basin (this study indicated by a yellow rectangle); 5. Qaidam Basin; 6. Xining Basin; 7. Jianchuan Basin; 8. Xigaze Basin; 9. Markam Basin; 10. Gonjo Basin.

[revised manuscript text omitted]
, 2017). In contrast, the westerlies were the chief source of precipitation on the northern TP (see Section 5.2) and would have carried cold-temperate conifer pollen from the mountains surrounding northeastern Tibet. Therefore, wBased on the above, we propose that both discrete local climatic conditions and the influence of different regional
atmospheric circulation systems were both primary driving factors ofcontributed to the development of a unique
distinct floral ecosystems in the northern and southeast-central TP Tibet during the middle late Eocene.

**6. Conclusions**

On the basis of palynological assemblages, we conclude that the rocks of the RZ section (Nangqian Basin)
are late Lutetian Bartonian (late middle 41.2–37.8 Ma; late Eocene) in age. They record a strongly seasonal
steppe-desert ecosystem characterised by *Ephedra* and *Nitraria* shrubs, diverse ferns and an underlying
component of broad-leaved forests. The climate became significantly warmer over the MECOfor a short period,
encouraging regional forest growth and a proliferation of the thermophilic ancestral *Ephedra* type, but rapidly
aridified thereafter due primarily to regression of the proto-Paratethys Sea. This is in conjunction with observed
environmental shifts in northeastern Tibet, and provides further support forsuggesting widespread Asian
aridification after 40 Main the late Eocene. A new palynozonation better constrains the biostratigraphy of
Paleogene successions across the northern, central, and southern TP, and also illustrates local ecological
variability during the Eocene. This highlights the ongoing challenge of integrating various deep time records for
the purpose of reconstructing palaeoelevation, and suggests that a multiproxy approach is vital for unravelling
the complex uplift history of the Qinghai-Tibetan regionTP.

**Author contribution**

Q.Y., V.V., F.S.S., D.L.G., H.C.W. and Q.S.F. conceptualized the study. Q.Y., F.S.S., H.C.W., Z.J.Q., Y.S.D.
and J.J.S. carried out fieldwork. Q.Y., N.B., V.V. and C.R. collected and analysed the data. Q.Y. wrote the first
draft and N.B., V.V., and C.R. participated in review and editing of the final draft.

**Competing interests**

The authors declare that they have no conflict of interest.

**Acknowledgments**

We thank Dr. Fuyuan An (Qinghai Normal University), Dr. Shuang Lü (University of Tübingen), and Aijun
Sun (University of Chinese Academy of Sciences) for assistance in the fieldwork, and Prof. Yunfa Miao (Chinese Academy of Sciences) for helpful discussions on the systematic palynology, and two anonymous reviewers for their constructive comments. This work was supported by the National Natural Science

Foundation of China (grant 41302024 to Q.Y.); The Youth Guiding Fund of Qinghai Institute of Salt Lakes,

CAS (grant Y360391053 to Q.Y.); The Second Tibetan Plateau Scientific Expedition and Research Program (STEP) CAS (grant 2019 QZKK0805 to Q.Y.), and the Swedish Research Council (VR) grants 2015-4264 to

V.V. and 2017-03985 to C.R. Funding sources had no involvement in study design.

**Data availability**

The authors declare that all data supporting the findings of this study are available in the supplementary information or published in a data repository at the following DOI: http://dx.doi.org/10.17632/xvp68wsd2p.4

10.17632/xvp68wsd2p.2.

**Supplementary information**

Supplementary information is available for this paper (Fig. S1, S2, S3).

[revised manuscript text omitted]

---

## Author Response (AR2)

October 2020

Editorial Board

Climate of the Past

Dear Prof. Alberto Reyes

Reply to reviewer comments on MS No.: cp-2019-138

We thank you for your time and effort in reviewing our manuscript "Aridification signatures from fossil pollen indicate a drying climate in east-central Tibet during the late Eocene". Below we provide responses to the comments with line numbers indicating where we have made changes, and include a separate marked-up manuscript version showing the changes made.

*\*Please be advised, line numbers below refer to lines in the marked-up version of the document, not the clean copy.*

Comments to the Author

*1. Yuan et al 2017 Paleoworld paper gives a slightly different account of the stratigraphic framework for the Gongjue Fm: the 2017 paper describes three members for the formation as a whole, but this manuscript gives those same three members within only the Eg3 unit of the formation. Please clarify for readers.*

Thank you for pointing this out. We realised that the stratigraphy described in Yuan et al. (2017, Paleoworld) was incorrect when checking the original papers in Chinese of Wang and Wang (2001, 2002) and Du et al. (2011). We have now made this clear for readers in Lines 162-165.

*2. I realize that authors are preparing a separate manuscript with details on lithostrat, geochemistry, etc for this section. But because there is no published information on this section it is fair to ask for at least some additional brief details. Something like Table 1 in the Yuan et al 2017 paper would be sufficient, along with some representative photographs similar to Fig 2 of that same paper. Are there any potential linkages between lithofacies in the upper part of the section and the poor sample productivity in Zones 2 and 3? If so, please add some short on this point, since the poor pollen recovery in those zones is certainly problematic (as the authors already point out).*

We now added a new figure (Fig. 2) illustrating the lithologies of the Ria Zhong (RZ) section. Field photographs of the section with palynological zones marked, and images of the representative lithologies are included. A new table (Table 1) has also been added, giving further details on the lithostratigraphy of the RZ section (Lines 191-202).

We added a discussion on the linkage between lithofacies in the upper part of the section and the poor sample productivity in Zones 2 and 3 in Lines 230-235.

*3. Somewhat related to the point above, I think it's fair to ask that authors be even more circumspect about the problematic low pollen recovery from sediments assigned to Zones 2 and 3. This could be done easily at the start of the Results (lines 214): "Recovery is particularly poor in the upper half (~xx m) of the section." This should then be followed up with caveats in the results section for those zones (section 4.1.2 and 4.1.3).*

We have now added these points in Lines 230, 332-334, and 340-341.

*4. Authors have noted at one point in the Discussion that the tropical forest "spike" is only present in one sample, but elsewhere in the Discussion (lines 372 and 374) this spike is assigned to the entire Zone 2. Please revise to make it crystal clear that this spike only applies to one sample. e.g ".....a significant spike in tropical forest pollen in one sample…" and "Although the increase in tropical forest taxa in one sample from this zone does not indicate…"*

We now added this text in Lines 244, 329, 401 and 403.

*5. Very minor point: on line 200 you say you followed the approach of Hoorn et al to assign taxa to PFTs. Can you clarify if you followed the taxon-to-NLR assignments provided by Hoorn? If there are any deviations from this, or if this is not what you meant, please provide a supplemental table that explicitly gives the NLR and PFT associated with each taxon. Hmm, on second thought the supplemental pollen diagrams probably accomplish this already for the PFTs, at least.*

Yes, we followed the taxon-to-NLR assignments of Hoorn et al. (2012) without deviations. PFTs are shown in the supplementary dataset of Yuan & Barbolini (2020) for which the DOI is provided in the paper, as well as Fig. S1 and S2. We now added this text for readers in Lines 215-217 so they can easily find the information should they wish to.

[revised manuscript text omitted]